# Optimal Batched Best Arm Identification

**Tianyuan Jin[1], Yu Yang[3], Jing Tang[2], Xiaokui Xiao[1], Pan Xu[3]**
[1]National University of Singapore
[2]The Hong Kong University of Science and Technology (Guangzhou)
[3]Duke University
{tianyuan,xkxiao}@nus.edu.sg, jingtang@ust.hk, {yu.yang,pan.xu}@duke.edu

## Abstract

We study the batched best arm identification (BBAI) problem, where the learner's goal is to identify the best arm while switching the policy as less as possible. In particular, we aim to find the best arm with probability $1-\delta$ for some small constant $\delta > 0$ while minimizing both the sample complexity (total number of arm pulls) and the batch complexity (total number of batches). We propose the three-batch best arm identification (Tri-BBAI) algorithm, which is the first batched algorithm that achieves the optimal sample complexity in the asymptotic setting (i.e., $\delta \to 0$) and runs in 3 batches in expectation. Based on Tri-BBAI, we further propose the almost optimal batched best arm identification (Opt-BBAI) algorithm, which is the first algorithm that achieves the near-optimal sample and batch complexity in the non-asymptotic setting (i.e., $\delta$ is finite), while enjoying the same batch and sample complexity as Tri-BBAI when $\delta$ tends to zero. Moreover, in the non-asymptotic setting, the complexity of previous batch algorithms is usually conditioned on the event that the best arm is returned (with a probability of at least $1 - \delta$), which is potentially unbounded in cases where a sub-optimal arm is returned. In contrast, the complexity of Opt-BBAI does not rely on such an event. This is achieved through a novel procedure that we design for checking whether the best arm is eliminated, which is of independent interest.

## 1 Introduction

Multi-armed bandit (MAB) is a fundamental model in various online decision-making problems, including medical trials [40], online advertisement [5], and crowdsourcing [43]. These problems typically involve a bandit with multiple arms, where each arm follows an unknown distribution with a mean value. At each time step, the learner selects an arm, and receives a reward sample drawn from the chosen arm's distribution. Best arm identification (BAI) aims to identify the arm with the highest mean reward, which can be approached with a fixed budget or a fixed confidence level [14, 4, 19].

In this paper, we assume that there is a unique best arm and study BAI with fixed confidence. Specifically, we consider a set $[n] = \{1, 2, \ldots, n\}$ of $n$ arms, where each arm $i$ is associated with a reward distribution having a mean value $\mu_i$. Without loss of generality, for a bandit instance denoted by $\boldsymbol{\mu} = \{\mu_1, \mu_2, \ldots, \mu_n\}$, we assume that $\mu_1 > \mu_2 \geq \cdots \geq \mu_n$. At each time step $t$, the learner selects an arm and observes a sample drawn independently from the chosen arm's distribution. In the fixed confidence setting, the learner aims to correctly identify the best arm (arm 1 in our context) with a probability of at least $1 - \delta$, where $\delta > 0$ is a pre-specified confidence parameter. Meanwhile, the learner seeks to minimize the total number of arm pulls, also known as the sample complexity.

Denote by $a^*(\boldsymbol{\lambda}) = \arg\max_i \lambda_i$ the best arm for an arbitrary bandit instance $\boldsymbol{\lambda}$. Let $\text{Alt}(\boldsymbol{\mu}) = \{\boldsymbol{\lambda} \colon a^*(\boldsymbol{\lambda}) \neq 1\}$ be a set of models that have a different best arm from the model $\boldsymbol{\mu}$, and $\mathcal{P}_k = \{\boldsymbol{w} \in \mathbb{R}_+^n \colon \sum_{i=1}^n w_i = 1\}$ be the probability simplex. Garivier and Kaufmann [17] showed that for bandits with reward distributions that are continuously parameterized by their means, the number $N_\delta$ of pulls

38th Conference on Neural Information Processing Systems (NeurIPS 2024).

by any algorithm that returns the best arm with a probability of at least $1 - \delta$ is bounded by

$$\liminf_{\delta \to 0} \frac{\mathbb{E}_{\boldsymbol{\mu}}[N_\delta]}{\log(1/\delta)} \geq T^*(\boldsymbol{\mu}), \tag{1.1}$$

where $T^*(\boldsymbol{\mu})$ is defined according to

$$T^*(\boldsymbol{\mu})^{-1} := \sup_{\boldsymbol{w} \in \mathcal{P}_k} \left( \inf_{\boldsymbol{\lambda} \in \text{Alt}(\boldsymbol{\mu})} \left( \sum_{i=1}^{n} w_i \cdot d(\mu_i, \lambda_i) \right) \right), \tag{1.2}$$

and $d(\mu_i, \lambda_i)$ is the Kullback-Leibler (KL) divergence of two arms' distributions with means $\mu_i$ and $\lambda_i$, respectively. We say that an algorithm achieves the asymptotically optimal sample complexity if it satisfies

$$\limsup_{\delta \to 0} \frac{\mathbb{E}_{\boldsymbol{\mu}}[N_\delta]}{\log(1/\delta)} \leq T^*(\boldsymbol{\mu}). \tag{1.3}$$

The well-known Track-and-Stop algorithm [17] solves the BAI problem with asymptotically optimal sample complexity. However, it is a fully sequential algorithm, which is hard to be implemented in parallel. The learner in such an algorithm receives immediate feedback for each arm pull, and adjusts the strategy for the next arm selection based on the previous observations. Unfortunately, this sequential approach may not be feasible in many real-world applications. For instance, in medical trials, there is typically a waiting time before the efficacy of drugs becomes observable, making it impossible to conduct all tests sequentially. Instead, the learner needs to group the drugs into batches and test them in parallel. Similarly, in crowdsourcing, the goal is to identify the most reliable worker for a specific task by testing candidates with a sequence of questions. Again, there is often a waiting time for workers to finish all the questions, necessitating the grouping of workers into batches and conducting parallel tests. In such scenarios, the results of pulling arms are only available at the end of each batch [22, 1, 39, 37, 15, 24, 25, 38].

Motivated by these applications, we study the problem of batched best arm identification (BBAI). In BBAI, we are allowed to pull multiple arms in a single batch, but the results of these pulls are revealed only after the completion of the batch. The objective is to output the best arm with a high probability of at least $1 - \delta$, while minimizing both the sample complexity (total number of pulls) and the batch complexity (total number of batches). This leads to the following natural question:

*Can we solve the BBAI problem with an asymptotically optimal sample complexity and only using a constant number of batches?*

Furthermore, the aforementioned results hold only in the limit as the confidence parameter $\delta$ approaches zero, which may provide limited practical guidance since we often specify a fixed confidence level parameter $\delta > 0$. To address this, some algorithms [31, 21] have been proposed to solve the BAI problem with finite confidence. Specifically, for some universal constant $C$, these algorithms satisfy that with probability $1 - \delta$,

$$\mathbb{E}[N_\delta] \leq O\left( \sum_{i>1} \frac{1}{\Delta_i^2} \log \log \Delta_i^{-1} \right) \tag{1.4}$$

where $\Delta_i = \mu_1 - \mu_i$. In addition, Jamieson et al. [21] demonstrated that for two-armed bandits, the term $\frac{\log \log \Delta_2^{-1}}{\Delta_2^2}$ is optimal as $\Delta_2 \to 0$, where $\Delta_2$ is assumed to be the gap between the best arm and the second best arm. In the context of the batched setting, Jin et al. [22] proposed an algorithm that achieves the sample complexity in (1.4) within $\mathcal{O}(\log^*(n) \cdot \log(1/\Delta_2))$ batches, where $\log^*(n)$ is the iterated logarithm function[1]. Furthermore, Tao et al. [39] proved that for certain bandit instances, any algorithm that achieves the sample complexity bound shown in (1.4) requires at least $\Omega\left( \frac{\log \Delta_2^{-1}}{\log \log \Delta_2^{-1}} \right)$ batches. It should be noted that the batched lower bound proposed by Tao et al. [39] assumes $\delta$ as a constant, making it inapplicable in the asymptotic setting. Therefore, an additional question that arises is:

*Can we achieve the optimal sample complexity in (1.3) and (1.4) adaptively, taking into account the specified confidence level parameter $\delta$, while minimizing the number of batches?*

---

[1]Specifically, $\log^*(n)$ denotes the number of times the function $\log(\cdot)$ needs to be applied to $n$ until the result is less than 1.

Table 1: Comparison of sample and batch complexity of different algorithms. In the asymptotic setting (i.e., $\delta \to 0$), the sample complexity of an algorithm is optimal if it satisfies the definition in (1.3). The field marked with "–" indicates that the result is not provided. The sample complexity presented for [31, 22] is conditioned on the event that the algorithm returns the best arm, which can be unbounded when it returns a sub-optimal arm with certain (non-zero) probability (see Remark 4.4 for more details). In contrast, the sample complexity presented for [17, 26] and our algorithms is the total expected number of pulls that will be executed.

| Algorithm | Asymptotic behavior ($\delta \to 0$) | | Finite-confidence behavior | |
|---|---|---|---|---|
| | Sample complexity | Batch complexity | Sample complexity | Batch complexity |
| Karnin et al. [31] | Not optimal | $\mathcal{O}(\log n \cdot \log \frac{n}{\Delta_2})$ | $\mathcal{O}\left(\sum_{i>1} \frac{\log(\log \Delta_i^{-1})}{\Delta_i^2}\right)$ | $\mathcal{O}(\log n \cdot \log \frac{n}{\Delta_2})$ |
| Jin et al. [22] | Not optimal | $\mathcal{O}(\log \frac{1}{\Delta_2})$ | $\mathcal{O}\left(\sum_{i>1} \frac{\log(\log \Delta_i^{-1})}{\Delta_i^2}\right)$ | $\mathcal{O}(\log^*(n) \cdot \log \frac{1}{\Delta_2})$ |
| Wang et al. [42] | Optimal | $O(\sqrt{\log(1/\delta)})$ | $O(nH(\boldsymbol{\mu})^4/w_{\min}^2)$ | - |
| Agrawal et al. [2] | Optimal | $\frac{T^*(\boldsymbol{\mu})\log \delta^{-1}}{m}$ $(m = o(\log \delta^{-1}))$ | – | – |
| Karpov et al. [32] | – | – | $\mathcal{O}\left(\sum_{i>1} \frac{\log(n \log \Delta_i^{-1})}{\Delta_i^2}\right)$ | $\mathcal{O}(\log \frac{1}{\Delta_2})$ |
| Lower bound [39] | – | – | $\mathcal{O}\left(\sum_{i>1} \frac{\log(\log \Delta_i^{-1})}{\Delta_i^2}\right)$ | $\Omega(\log(1/\Delta_2))$ |
| **Tri-BBAI** (Our Algorithm 1) | Optimal | 3 | – | – |
| **Opt-BBAI** (Our Algorithm 2) | Optimal | 3 | $\mathcal{O}\left(\sum_{i>1} \frac{\log(n \log \Delta_i^{-1})}{\Delta_i^2}\right)$ | $\mathcal{O}(\log \frac{1}{\Delta_2})$ |

In this work, we provide positive answers to both of the aforementioned questions. Specifically, the **main contributions** of our paper can be summarized as follows:

- We propose Tri-BBAI (Algorithm 1) that returns the best arm with a probability of at least $1 - \delta$ by pulling all arms for a total number of at most $T^*(\boldsymbol{\mu}) \log(1/\delta)$ times when $\delta \to 0$. Tri-BBAI employs three batches in expectation when $\delta \to 0$. Therefore, Tri-BBAI achieves the optimal sample within constant batches. As a comparison, Track-and-Stop [17] also achieves the optimal sample complexity but requires solving the right-hand side of (1.2) after each arm pull, resulting in a batch complexity of the same order as the sample complexity, which is a significant computational overhead in practice.

- Built upon Tri-BBAI, we further propose Opt-BBAI (Algorithm 2) that runs in $\mathcal{O}(\log(1/\Delta_2))$ expected batches and pulls at most $\mathcal{O}\left(\sum_{i>1} \Delta_i^{-2} \log(n \log \Delta_i^{-1})\right)$ expected number of arms for finite confidence $\delta$. It is also important to note that Opt-BBAI achieves the same sample and expected batch complexity as Tri-BBAI asymptotically when $\delta \to 0$. Moreover, for the finite confidence case, this sample complexity matches (1.4) within a $\log(\cdot)$ factor and matches the optimal batched complexity within a $\log \log(\cdot)$ factor. To the best of our knowledge, Opt-BBAI is the first batched bandit algorithm that can achieve the optimal asymptotic sample complexity and the near-optimal non-asymptotic[2] sample complexity adaptively based on the assumption on $\delta$.

- Notably, in the non-asymptotic setting, the complexity of earlier batch algorithms [22, 32, 39, 24] typically depends on the event of returning the best arm, which occurs with a probability of at least $1 - \delta$. However, this complexity could potentially become unbounded if a sub-optimal arm is returned instead. Unlike these algorithms, the complexity of Opt-BBAI is not contingent on such an event. This is made by employing an innovative procedure to verify if the best arm is eliminated, a method that holds its independent interest. To the best of our knowledge, Opt-BBAI is the first algorithm that achieves the optimal asymptotic sample complexity while providing the optimal non-asymptotic sample complexity within logarithm factors.

- We also conduct numerical experiments[3] to compare our proposed algorithms with the optimal sequential algorithm Track-and-Stop [17], and the batched algorithm Top-k $\delta$-Elimination [22] on various problem instances. The results indicate that our algorithm significantly outperforms the Track and Stop method in terms of batch complexity, while its sample complexity is not much worse than that of Track and Stop. Additionally, our algorithm demonstrates a notable improvement in sample complexity compared to [22], while exhibiting similar batch complexity.

For ease of reference, we compare our results with existing work on batched bandits in Table 1.

---

[2]Non-asymptotic here refers to finite confidence. In this paper, we use names non-asymptotic and finite confidence interchangeably.

[3]Due to the space limit, we put the experimental results in Appendix E.

Table 2: Comparison of sample complexity of different algorithms.

| Algorithm | Asymptotic behavior ($\delta \to 0$) | Finite-confidence behavior |
|---|---|---|
| Kalyanakrishnan et al. [30] | $O(H(\boldsymbol{\mu})\log(1/\delta))$ | $O(H(\boldsymbol{\mu})\log(H(\boldsymbol{\mu})))$ |
| Karnin et al. [31] | $O(H(\boldsymbol{\mu})\log(1/\delta))$ | $\mathcal{O}\left(\sum_{i>1}\frac{\log(\log \Delta_i^{-1})}{\Delta_i^2}\right)$ |
| Garivier and Kaufmann [17] | $T^*(\boldsymbol{\mu})\log(1/\delta)+o(1/\delta)$ | - |
| Jamieson et al. [21] | $O(H(\boldsymbol{\mu})\log(1/\delta))$ | $\mathcal{O}\left(\sum_{i>1}\frac{\log(\log \Delta_i^{-1})}{\Delta_i^2}\right)$ |
| Jourdan and Degenne [26] | $T^*_\beta(\boldsymbol{\mu})\log(1/\delta)+o(1/\delta)$ | $\mathcal{O}\left((H(\boldsymbol{\mu})\log H(\boldsymbol{\mu}))^\alpha\right), \alpha>1$ |
| Degenne et al. [12] | $T^*(\boldsymbol{\mu})\log(1/\delta)+o(1/\delta)$ | $\tilde{O}(nT^*(\boldsymbol{\mu})^2)$ |
| Katz-Samuels et al. [33] | $O(T^*(\boldsymbol{\mu})\log(1/\delta))$ | $O(H(\boldsymbol{\mu})\log(n/\Delta_{\min}))$ |
| Wang et al. [42] | $T^*(\boldsymbol{\mu})\log(1/\delta)+o(1/\delta)$ | $O(nH(\boldsymbol{\mu})^4/w_{\min}^2)$ |
| **Opt-BBAI** (Our Algorithm 2) | $T^*(\boldsymbol{\mu})\log(1/\delta)+o(1/\delta)$ | $\mathcal{O}\left(\sum_{i>1}\frac{\log(n\log \Delta_i^{-1})}{\Delta_i^2}\right)$ |

## 2 Related Work

The BAI problem with fixed confidence is first studied for $[0,1]$ bounded rewards by Even-Dar et al. [14]. The sample complexity of their algorithm scales with the sum of the squared inverse gap, i.e., $H(\boldsymbol{\mu}) = \sum_{i>1} 1/\Delta_i^2$. Garivier and Kaufmann [17] showed that $H(\boldsymbol{\mu}) < T^*(\boldsymbol{\mu}) \le 2H(\boldsymbol{\mu})$ with $T^*(\boldsymbol{\mu})$ defined in (1.2) and proposed the Track-and-Stop algorithm, which is the first one in the literature proved to be asymptotically optimal. Later, Degenne et al. [12] viewed $T^*(\boldsymbol{\mu})$ as a min-max game and provided an efficient algorithm to solve it. Jourdan et al. [27] studied the asymptotically optimal sample complexity for any reward distributions with bounded support. Degenne et al. [13] studied pure exploration in linear bandits. Their proposed algorithm proved to be both asymptotically optimal and empirically efficient.

There are also many studies [30, 10, 8, 21, 31] that focus on providing non-asymptotic optimal sample complexity. The best non-asymptotic sample complexity was achieved by Chen et al. [10], which replaces the term $\log\log \Delta_i^{-1}$ in (1.4) with a much smaller term. Furthermore, when we allow a loss $\epsilon$ of the quality of the returned arm, the problem is known as $(\epsilon, \delta)$-PAC BAI, for which various algorithms [29, 14, 6, 9] are proposed, achieving the worst-case optimal sample complexity.

There are a few attempts to achieve both the asymptotic and non-asymptotic optimal sample complexity. Degenne et al. [12] provided a non-asymptotic sample complexity $\tilde{O}(nT^*(\boldsymbol{\mu})^2)$, which could be $nT^*(\boldsymbol{\mu})$ larger than the optimal sample complexity. Recently, Jourdan and Degenne [26] managed to achieve a sample complexity that is $\beta$-asymptotically optimal (with $w_1$ fixed at $1/\beta$ in (1.2)), rendering it asymptotically optimal up to a factor of $1/\beta$. Meanwhile, they also reached a non-asymptotic sample complexity of $\mathcal{O}\big((H(\boldsymbol{\mu}) \cdot \log H(\boldsymbol{\mu}))^\alpha\big)$[4] for some $\alpha > 1$, where $H(\boldsymbol{\mu}) = \sum_{i>1} 1/\Delta_i^2$. Wang et al. [42] explored both asymptotic and non-asymptotic sample complexities. Their algorithm achieves asymptotic optimality and shows a non-asymptotic sample complexity of $O(nH(\boldsymbol{\mu})^4/w_{\min}^2)$. However, this non-asymptotic sample complexity is $nH(\boldsymbol{\mu})^3/w_{\min}^2$ away from being optimal. Jourdan et al. [28] studied $(\epsilon, \delta)$-PAC BAI, proposing an asymptotically optimal algorithm and providing non-asymptotic sample complexity. When $\epsilon = 0$, it aligns with our setting, our non-asymptotic sample complexity is better scaled. Specifically, Jourdan et al. [28] offered a non-asymptotic sample complexity scale of $n/\Delta_2^2 \log(1/\Delta_2)$, whereas ours is more instance-sensitive, as our sample complexity is related to all gaps, not just $\Delta_2$. Additionally, Jourdan et al. [28] considered a practical scenario where the algorithm can return a result at any time while still ensuring a good guarantee on the returned arm. For ease of reference, we summarize the sample complexity of different algorithms in Table 2. As shown in Table 2, our algorithm is the only one that achieves both the asymptotic optimality and non-asymptotic optimality within logarithm factors.

Another line of research [31, 4, 7] investigated BAI with a fixed budget, where the objective is to determine the best arm with the highest probability within $T$ pulls. Audibert et al. [4] and Karnin et al. [31] offered finite-time bounds for this problem, while Carpentier and Locatelli [7] presented a tight lower bound, demonstrating that such finite-time bounds[4, 31] are optimal for certain bandit

---

[4]To derive the near-optimal non-asymptotic sample complexity, $\alpha$ should be 1. However, As explained in their original paper, the algorithm will be sub-optimal if we set $\alpha$ close to 1.

instances. However, the asymptotic optimality for this problem remains unknown. Interested readers are referred to recent advancements[11, 35] in the asymptotic results of BAI with a fixed budget.

In addition, some recent works [1, 22, 39, 32] also focused on batched BAI in non-asymptotic setting. Agarwal et al. [1] studied the batched BAI problem under the assumption that $\Delta_2$ is known. They proposed an algorithm that has the worst-case optimal sample complexity of $\mathcal{O}(n\Delta_2^{-2})$ and runs in $\log^*(n)$ batches. Later, Jin et al. [22] provided the algorithms that achieves the sample complexity given in (1.4) within $\tilde{\mathcal{O}}(\log(1/\Delta_2))$ batches, where $\tilde{O}$ hides the $\log\log(\cdot)$ factors. Tao et al. [39] studied the BAI problem in the general collaborative setting and showed that no algorithm can achieve (1.4) with $o((\log \Delta_2^{-1})/\log\log \Delta_2^{-1})$ batches. Karpov et al. [32] further proposed an algorithm which has the sample complexity $\sum_{i>1} \frac{\log(n\log \Delta^{-1})}{\Delta_i^2}$ and the batch complexity $O(\log(1/\Delta_2))$. We note that the lower bound of batch complexity given by Tao et al. [39] can only be applied to a constant $\delta$. In other words, the lower bound of complexity for the asymptotic setting remains unknown. Agrawal et al. [2] studied the optimal batch size for keeping the asymptotic optimality. They showed an algorithm with batch size $m = o(\log(1/\delta))$ achieves the asymptotic optimality. The batch complexity of the algorithm is $O(T^*(\boldsymbol{\mu})\log(1/\delta)/m)$. Wang et al. [42] provided an algorithm that uses $O(\sqrt{\log \delta^{-1}})$ batches and retains asymptotic optimality for linear bandits. However, for the asymptotic setting, such batch size is still too large as it grows to infinity as $\delta$ decreases to 0.

# 3 Achieving Asymptotic Optimality with at Most Three Batches

## 3.1 Reward Distribution

We assume the reward distributions belong to a known one-parameter exponential family that is commonly considered in the literature [17, 16, 36]. In particular, the measure $\nu_\theta$ of such probability distributions with respect the model parameter $\theta$ satisfies $\frac{d\nu_\theta}{d\rho}(x) = \exp(x\theta - b(\theta))$, for some measure $\rho$ and $b(\theta) = \log(\int e^{x\theta}d\rho(x))$. For one-parameter exponential distributions, it is known that $b'(\theta) = \mathbb{E}[\nu_\theta]$ and the mapping $b'(\theta) \mapsto \theta$ is one-to-one. Moreover, given any two mean values $\mu$ and $\mu'$, we define $d(\mu, \mu')$ to be the Kullback-Leibler divergence between two distributions with mean values $\mu$ and $\mu'$.

## 3.2 The Proposed Three-Batch Algorithm

Algorithm 1 shows the pseudo-code of our proposed Tri-BBAI algorithm. In particular, Tri-BBAI has four stages. In what follows, we elaborate on the details of each stage.

**Stage I: Initial exploration.** In this stage, we pull each arm for $L_1$ times. Denote by $i^*(t)$ the arm with the largest empirical mean at time $t$ (i.e., after we pull all arms for a total number of $t$ times), i.e., $i^*(t) = \max_{i\in[n]} \hat{\mu}_i(t)$. Let $\tau_0 = nL_1$. Fix $t = \tau_{q-1} \geq \tau_0$, we let $b_i^q = \hat{\mu}_i(t) + \epsilon$ for $i \neq i^*(t)$ and $b_i^q = \hat{\mu}_i(t) - \epsilon$ for $i = i^*(t)$. Let $\boldsymbol{w}^*(\boldsymbol{\mu}) = \arg\max_{\boldsymbol{w}\in\mathcal{P}_k} \inf_{\boldsymbol{\lambda}\in\text{Alt}(\boldsymbol{\mu})} \left(\sum_{i=1}^n w_i \cdot d(\mu_i, \lambda_i)\right)$. Then, for the aforementioned $\boldsymbol{b}^q = \{b_1^q, b_2^q, \ldots, b_n^q\}$, we calculate $\boldsymbol{w}^*(\boldsymbol{b}^q)$ according to Lemma A.1 and $T^*(\boldsymbol{b}^q)$ according to Lemma A.2. We note that arm 1 is assumed to be the arm with the highest mean in these two lemmas. However, in the context of $\boldsymbol{b}^q$, the index of $i^*(t)$ might be different. To align with the standard practice, we can rearrange the indices of the arms in $\boldsymbol{b}^q$ so that $i^*(t)$ corresponds to index 1.

**Purpose.** To achieve the asymptotic optimality, we attempt to pull each arm $i$ for around $w_i^*(\boldsymbol{\mu})T^*(\boldsymbol{\mu})$ times. We can show that with a high probability, $w_i^*(\boldsymbol{b}^q)T^*(\boldsymbol{b}^q)$ is close to $w_i^*(\boldsymbol{\mu})T^*(\boldsymbol{\mu})$, which implies that pulling arm $i$ for a number of times proportional to $w_i^*(\boldsymbol{b}^q)T^*(\boldsymbol{b}^q)$ is likely to ensure the asymptotic optimal sample complexity.

**Stage II: Exploration using $\boldsymbol{w}^*(\boldsymbol{b}^q)$ and $T^*(\boldsymbol{b}^q)$.** Stage II operates in batches with the maximum number of batches determined by $\log(1/\delta)$. At batch $q$, each arm $i$ is pulled $\max_{p:p\in\mathbb{N},p\in[1,q]} T_i^q$ times in total. Here

$$T_i^q := \min\left\{\alpha w_i^*(\boldsymbol{b}^q)T^*(\boldsymbol{b}^q)\log \delta^{-1}, L_2\right\}, \tag{3.1}$$

where the definition of $w_i^*(\boldsymbol{b}^q)$ and $T^*(\boldsymbol{b}^q)$ could be found in Stage I. We then evaluate the stage switching condition, $|w_i^*(\boldsymbol{b}^q) - w_i^*(\boldsymbol{b}^{q-1})| \leq 1/\sqrt{n}$ for all $i \in [n]$. If this condition is met, we go to the next stage; otherwise, we proceed to the next batch within Stage II.

---

**Algorithm 1:** Three-Batch Best Arm Identification (Tri-BBAI)

---

**Input:** Parameters $\epsilon, \delta, L_1, L_2, L_3, \alpha$ and function $\beta(t, \delta)$.
**Output:** The estimated optimal arm.

1  *Stage I: Round 1 exploration*
2  **for** $i \leftarrow 1$ *to* $n$ **do**
3      Pull arm $i$ for $L_1$ times;

4  $t \leftarrow nL_1$ and $\tau_0 \leftarrow t$;
5  *Stage II:Round 2 exploration*
6  Let $w_0^*(\boldsymbol{b}^0) = (1/n, 1/n, \cdots, 1/n)$ and $T_i^0 = L_1$ for all $i \in [n]$;
7  **for** $q = 1, 2 \cdots, \log(1/\delta)$ **do**
8      $i^*(t) \leftarrow \max_{i \in [n]} \hat{\mu}_i(t)$;
9      **for** *each* $i \in [n] \setminus \{i^*(t)\}$ **do**
10        $b_i^q \leftarrow \hat{\mu}_i(t) + \epsilon$;
11     $b_{i^*(t)}^q \leftarrow \hat{\mu}_{i^*(t)}(t) - \epsilon$;
12     **for** $i \leftarrow 1$ *to* $n$ **do**
13        Pull arm $i$ for $\{0, T_i^q - \max_{p:p \in \mathbb{N}, p \in [0,q)} T_i^p\}$ times;
          /**/ Note that $T_i^q = \min\left\{\alpha w_i^*(\boldsymbol{b}^q)T^*(\boldsymbol{b}^q)\log \delta^{-1}, L_2\right\}$ by (3.1)
14        $t \leftarrow t + \{0, T_i^q - \max_{p:p \in \mathbb{N}, p \in [0,q)} T_i^p\}$;
15     **if** $|w_i^*(\boldsymbol{b}^q) - w_i^*(\boldsymbol{b}^{q-1})| \leq 1/\sqrt{n}$ *for all* $i \in [n]$ **then**
16        Break;
17     $\tau_q \leftarrow t$;

18 $\tau \leftarrow t$, and $i^*(\tau) = \max_{i \in [n]} \hat{\mu}_i(\tau)$;
19 *Stage III: Statistical test with Chernoff's stopping rule*;
20 Compute $Z_j(\tau)$ according to (3.3) ;
21 **if** $\min_{j \in [n] \setminus \{i^*(\tau)\}} Z_j(\tau) \geq \beta(\tau, \delta/2)$ **then**
22     **return** $i^*(\tau)$;

23 *Stage IV: Round 3 exploration* **for** $i \leftarrow 1$ *to* $n$ **do**
24     Pull arm $i$ for total $\max\{0, L_3 - L_1 - T_i\}$ times;

25 $t \leftarrow nL_3$, and $i^*(t) \leftarrow \max_{i \in [n]} \hat{\mu}_i(t)$;
26 **return** $i^*(t)$ ;

---

**Purpose.** Pulling arm $i$ proportional to $w_i^*(\boldsymbol{b}^q)T^*(\boldsymbol{b}^q)$ provides statistical evidence for the reward distributions without sacrificing sample complexity compared to the optimality per our above discussion. Meanwhile, we also set a threshold $L_2$ to avoid over-exploration due to sampling errors from Stage I.

The rationale for running Stage II in multiple batches is based on empirical considerations. In experiments, $\delta$ is always finite. Consequently, the error of arm $i$, $|\hat{\mu}_i(t) - \mu_i|$, remains constant since the number of pulls of arms is limited. Given that the stopping rule in Stage III is highly dependent on the error of arm $i$ and the number of pulls $T_i := \max_{p:p \in \mathbb{N}, p \geq 1} T_i^p$, there is a constant probability that the stopping rule may not be met, leading to significant sample costs in Stage IV. Adding the condition in Line 15 ensures that $w^*(\boldsymbol{b}^q)$ converges and that the sample size $T_i^q$, as defined by $w^*(\boldsymbol{b}^q)$ and $T^*(\boldsymbol{b}^q)$, closely approximates $\alpha w^*(\boldsymbol{\mu})T^*(\boldsymbol{\mu})\log \delta^{-1}$. This alignment significantly increases the probability that the stopping rule will be satisfied in experiments.

Moreover, such modification doesn't hurt any theoretical results. To explain, in our analysis for Tri-BBAI, we demonstrate that as $\delta$ approaches 0, $w^*(\boldsymbol{b}^q)$ will be very close to $w^*(\boldsymbol{\mu})$ and the probability that Line 15 is not satisfied could be bounded by $1/\log^2 \delta^{-1}$, which means with high probability Stage II costs 2 batches. Besides, $q \leq \log(1/\delta)$, which implies that even if Line 15 is not satisfied, the number of batches required for Stage II is at most $\log(1/\delta)$. Therefore, the expected number of batches required for Stage II is 2 as $\delta$ approaches 0.

**Stage III: Statistical test with Chernoff's stopping rule.** Denote by $N_i(t)$ the number of pulls of arm $i$ at time $t$. For each pair of arms $i$ and $j$, define their weighted empirical mean as

$$\hat{\mu}_{ij}(t) := \frac{N_i(t) \cdot \hat{\mu}_i(t)}{N_i(t) + N_j(t)} + \frac{N_j(t) \cdot \hat{\mu}_j(t)}{N_i(t) + N_j(t)}, \tag{3.2}$$

where $\hat{\mu}_i(t)$ and $\hat{\mu}_j(t)$ are the empirical means of arms $i$ and $j$ at time $t$. For $\hat{\mu}_i(t) \geq \hat{\mu}_j(t)$, define

$$\begin{aligned} Z_{ij}(t) &:= d(\hat{\mu}_i(t), \hat{\mu}_{ij}(t))N_i(t) + d(\hat{\mu}_j(t), \hat{\mu}_{ij}(t))N_j(t), \\ Z_j(t) &:= Z_{i^*(t)j}(t). \end{aligned} \tag{3.3}$$

We test whether Chernoff's stopping condition is met (Line 21). If so, we return the arm with the largest empirical mean, i.e., $i^*(\tau)$, where $\tau$ is the total number of pulls examined after Stage II.

**Purpose.** The intuition of using Chernoff's stopping rule for the statistical test is two-fold. Firstly, if Chernoff's stopping condition is met, with a probability of at least $1 - \delta/2$, the returned arm $i^*(\tau)$ in Line 22 is the optimal arm (see Lemma B.1). Secondly, when $\delta$ is sufficiently small, with high probability, Chernoff's stopping condition holds (see Lemma B.4). As a consequence, our algorithm identifies the best arm successfully with a high probability of meeting the requirement.

**Stage IV: Re-exploration.** If the previous Chernoff's stopping condition is not met, we pull each arm until the total number of pulls of each arm eqauls $L_3$ taking into account the pulls in the previous stages (Line 24). Finally, the arm with the largest empirical mean is returned (Line 26).

**Purpose.** If Chernoff's stopping condition is not met, $i^*(\tau)$ might not be the optimal arm. In addition, when each arm is pulled for $L_3$ times, we are sufficiently confident that $i^*(t)$ is the best arm. Since the probability of Stage IV happening is very small, its impact on the sample complexity is negligible.

### 3.3 Theoretical Guarantees of Tri-BBAI

In the following, we present the theoretical results for Algorithm 1.

**Theorem 3.1** (Asymptotic Sample Complexity). Given any $\delta > 0$, let $\epsilon = \frac{1}{\log\log(\delta^{-1})}$, $L_1 = \sqrt{\log \delta^{-1}}$, $L_2 = \frac{\log \delta^{-1} \log\log \delta^{-1}}{n}$, and $L_3 = (\log \delta^{-1})^2$. Meanwhile, for any given $\alpha \in (1, e/2]$, define function $\beta(t, \delta)$ as $\beta(t, \delta) = \log(\log(1/\delta)t^\alpha/\delta)$.[5] Then, for any bandit instance $\boldsymbol{\mu}$, Algorithm 1 satisfies

$$\limsup_{\delta \to 0} \frac{\mathbb{E}_{\boldsymbol{\mu}}[N_\delta]}{\log(1/\delta)} \leq \alpha T^*(\boldsymbol{\mu}).$$

By letting $\alpha$ in Theorem 3.1 approach 1 (e.g., $\alpha = 1 + 1/\log \delta^{-1}$), we obtain the asymptotic optimal sample complexity.

**Theorem 3.2** (Correctness). Let $\epsilon$, $L_1$, $L_2$, $L_3$, $\alpha$, and $\beta(t, \delta)$ be the same as in Theorem 3.1. Then, for sufficiently small $\delta > 0$, Algorithm 1 satisfies $\mathbb{P}_{\boldsymbol{\mu}}(i^*(N_\delta) \neq 1) \leq \delta$.

**Theorem 3.3** (Asumptotic Batch Complexity). Let $\epsilon$, $L_1$, $L_2$, $L_3$, $\alpha$, and $\beta(t, \delta)$ be the same as in Theorem 3.1. For sufficiently small $\delta > 0$, Algorithm 1 runs within $3 + o(1)$ batches in expectation. Besides, Algorithm 1 runs within 3 batches with probability $1 - 1/\log(1/\delta^2)$.

To the best of our knowledge, all previous works in the BAI literature [18, 12, 2, 42] that achieve the asymptotic optimal sample complexity require unbounded batches as $\delta \to 0$. In contrast, Tri-BBAI achieves the asymptotic optimal sample complexity and runs within 3 batches in expectation, which is a significant improvement in the batched bandit setting where switching to new policies is expensive.

**Remark 3.4.** Apart from best arm identification, regret minimization is another popular task in bandits, where the aim is to maximize the total reward in $T$ rounds. Jin et al. [25] proposed an algorithm that achieves a constant batch complexity for regret minimization and showed that their algorithm is optimal when $T$ goes to infinity. In regret minimization, the cost of pulling the optimal arm is 0, indicating that the allocation $w_i$ (i.e., the proportion of pulling the optimal arm) is close to 1. In the BAI problem, the main hardness is to find the allocation $w_i$ for each arm since even pulling arm 1 will increase the sample complexity of the algorithm. Therefore, the strategy proposed by Jin et al. [25] cannot be applied to the BAI problem.

---

[5] Recent work by Kaufmann and Koolen [34] offers improved deviation inequalities allowing for a smaller selection of $\beta(t, \delta)$ without sacrificing the asymptotic optimality. However, it remains uncertain whether this refined parameter choice is applicable to our batched bandit problem. For ease of presentation, we use the parameter choice of $\beta$ in Garivier and Kaufmann [17].

---

**Algorithm 2:** (Almost) Optimal Batched Best Arm Identification (Opt-BBAI)

---

**Input:** Parameters $\delta, \epsilon, L_1, L_2, L_3, \alpha$ and function $\beta(t, \delta)$.

**Output:** The estimated optimal arm.

1  *Stage I, II, and III:* the same as that in Algorithm 1
2  *Stage IV: Exploration with Exponential Gap Elimination*
3  $S_r \leftarrow [n]$, $B_0 \leftarrow 0$, $r \leftarrow 1$;
4  **while** $|S_r| > 1$ **do**
5      Let $\epsilon_r \leftarrow 2^{-r}/4$, $\delta_r \leftarrow \delta/(40\pi^2 n \cdot r^2)$, $\ell_r \leftarrow 0$, and $\gamma_r \leftarrow \delta_r$;
6      *Successive elimination* /\*\*/ Eliminate arms whose means are lower than $\mu_1$ by at least $\epsilon_r$
7      **for** *each arm* $i \in S_r$ **do**
8          Pull arm $i$ for $d_r \leftarrow \frac{32}{\epsilon_r^2} \log(2/\delta_r)$ times;
9          Let $\hat{p}_i^r$ be the empirical mean of arm $i$;
10     Let $* \leftarrow \max_{i \in S_r} \hat{p}_i^r$;
11     Set $S_{r+1} \leftarrow S_r \setminus \{i \in S_r : \hat{p}_i^r < \hat{p}_*^r - \epsilon_r\}$;
12     *Checking for best arm elimination* /\*\*/ Reduce the risk of the best arm being eliminated
13     Let $B_r \leftarrow B_{r-1} + d_r |S_r|$;
14     **for** $j < r$ **do**
15         **for** *each arm* $i \in S_j \setminus S_{j+1}$ **do**
16             **if** $B_r \gamma_j \cdot 2^{\ell_j} > B_j$ **then**
17                 $\gamma_j \leftarrow (\gamma_j)^2$;
18                 Repull arm $i$ for total $\frac{32}{\epsilon_j^2} \log(2/\gamma_j)$ times;
19                 Let $\hat{p}_i^j$ be the empirical mean of arm $i$ in $S_j$;
20                 $\ell_j \leftarrow \ell_j + 1$;
21         **if** $\exists i \in S_j$, $\hat{p}_i^j > \hat{p}_*^r - \epsilon_j/2$ **then**
22             **return** Randomly return an arm in $S_r$;
23     $r \leftarrow r + 1$;
24 **return** The arm in $S_r$;

---

## 4 Best of Both Worlds: Achieving Asymptotic and Non-asymptotic Optimalities

The Tri-BBAI algorithm is shown to enjoy the optimal sample complexity with only three batches in the asymptotic setting. However, in practice, we are limited to a finite number of samples and thus $\delta$ cannot go to zero, which is a critical concern in real-world applications. Consequently, obtaining the optimal sample and batch complexity in a non-asymptotic setting becomes the ultimate objective of a practical bandit strategy in BBAI. In this section, we introduce Opt-BBAI, which can attain the optimal sample and batch complexity in asymptotic settings and near-optimal sample and batch complexity in non-asymptotic settings.

We assume a bounded reward distribution within $[0, 1]$, which aligns with the same setting in the literature [31, 21]. Again, we consider that the reward distribution belongs to a one-parameter exponential family. By refining Stage IV of Algorithm 1, we can achieve asymptotic optimality and near non-asymptotic optimality adaptively based on the assumption on $\delta$ in various settings.

The pseudo-code for the algorithm is provided in Algorithm 2. The main modification from Algorithm 1 occurs in Stage IV. Intuitively, if the algorithm cannot return at Stage III, then the value of $\log \delta^{-1}$ may be comparable to other problem parameters, such as $1/\Delta_2$. Therefore, we aim to achieve the best possible sample and batch complexity for the non-asymptotic scenario. Stage IV operates in rounds, progressively eliminating sub-optimal arms until a result is obtained. Each round consists of two components: *Successive Elimination* and *Checking for Best Arm Elimination*.

**Successive Elimination.** In the $r$-th round, we maintain a set $S_r$, a potential set for the best arm. Each arm in $S_r$ is then pulled $d_r = 32/\epsilon_r^2 \log(2/\delta_r)$ times. At Line 11, all possible sub-optimal arms are eliminated. This first component of Stage IV borrows its idea from successive elimination [14].

**Purpose.** After $d_r$ number of pulls, arms are likely to be concentrated on their true means within a distance of $\epsilon_r/4$ with a high probability. Hence, with high probability, the best arm is never eliminated at every round of Line 11, and the final remaining arm is the best arm.

*Issue of Best Arm Elimination.* Due to successive elimination, there is a small probability ($\leq \delta_r$) that the best arm will be eliminated. The following example illustrates that, conditioning on this small-probability event, the sample and batch complexity of the algorithm could become infinite.

**Example 4.1.** Consider a bandit instance where $\mu_1 > \mu_2 = \mu_3 > \mu_4 \geq \cdots \geq \mu_n$. If the best arm is eliminated at a certain round and never pulled again, the algorithm tasked with distinguishing between the 2nd and 3rd best arms will likely never terminate due to their equal means, leading to unbounded sample and batch complexity.

To address this issue, we introduce a *Check for Best Arm Elimination* component into Stage IV.

**Checking for Best Arm Elimination.** In the $r$-th round, we represent the total sample complexity used up to the $r$-th round as $B_r$. We employ $\gamma_j$ as an upper bound for the probability that the best arm is eliminated in $S_j$. If Line 16 is true ($B_r \gamma_j \cdot 2^{\ell_j} > B_j$), we adjust $\gamma_j$ to $\gamma_j^2$ and pull each arm in $S_j$ for $32/\epsilon_j^2 \log(2/\gamma_j)$ times (Line 18), subsequently updating their empirical mean (Line 19). Finally, we return a random arm in $S_r$ if the condition at Line 21 holds.

**Purpose.** If Line 16 is satisfied, it indicates that the sample costs, based on the event of the best arm being eliminated in the $r$-th round, exceed $B_j/2^{\ell_j}$. In this case, we increase the number of samples for arms in $S_j$ and re-evaluate if their empirical mean is lower than that of the current best arm $*$. This ensures that the expected total sample costs, assuming the best arm is eliminated at $S_j$, are bounded by $\sum_{\ell_j=1}^{\infty} B_j/2^{\ell_j} \leq B_j$. If Line 21 holds, we randomly return an arm from $S_r$. Since this only happens with a small probability, we still guarantee that Algorithm 2 will return the best arm with a probability of at least $1 - \delta$.

In what follows, we provide the theoretical results for Algorithm 2.

**Theorem 4.2.** Let $\epsilon$, $L_1$, $L_2$, $L_3$, $\alpha$, and $\beta(t, \delta)$ be the same as in Theorem 3.1. For finite $\delta \in (0, 1)$, Algorithm 2 identifies the optimal arm with probability at least $1 - \delta$ and there exists some universal constant $C$ such that $\mathbb{E}[N_\delta] \leq C\big(\sum_{i>1} \Delta_i^{-2} \log\big(n \cdot \log \Delta_i^{-1}\big)\big)$, and the algorithm runs in $C \log(1/\Delta_2)$ expected batches.

When $\delta$ is allowed to go to zero, we also have the following result.

**Theorem 4.3.** Let $\epsilon$, $L_1$, $L_2$, $L_3$, $\alpha$, and $\beta(t, \delta)$ be the same as in Theorem 3.1. Algorithm 2 identifies the optimal with probability at least $1 - \delta$ and its sample complexity satisfies $\limsup_{\delta \to 0} \mathbb{E}_{\boldsymbol{\mu}}[N_\delta]/\log(1/\delta) \leq \alpha T^*(\boldsymbol{\mu})$, and the expected batch complexity of Algorithm 2 converges to 3 when $\delta$ approaches 0.

The results in Theorems 4.2 and 4.3 state that Algorithm 2 achieves both the asymptotic optimal sample complexity and a constant batch complexity. Moreover, it also demonstrates near-optimal performance in both non-asymptotic sample complexity and batch complexity. Notably, this is the first algorithm that successfully attains the optimal or near-optimal sample and batch complexity, adaptively, in asymptotic and non-asymptotic settings.

**Remark 4.4.** Jin et al. [22] achieved near-optimal sample and batch complexity in a non-asymptotic setting. However, their results are contingent on the event that the algorithm can find the best arm (with probability $1 - \delta$). Consequently, with a probability of $\delta$ there is no guarantee for its batch and sample complexity to be bounded. As demonstrated in Example 4.1, the batch and sample complexity in [22] could even be infinite.

In contrast, the batch and sample complexity introduced in Theorem 4.2 is near-optimal and does not rely on any specific event due to the procedure "**checking for best arm elimination**" we proposed. Our technique could be of independent interest and could be further applied to existing elimination-based BAI algorithms [14, 21, 22, 31] to ensure that the sample and batch complexity is independent of the low-probability event that the best arm is eliminated.

# 5    Conclusion, Limitations, and Future Work

In this paper, we studied the BAI problem in the batched bandit setting. We proposed a novel algorithm, Tri-BBAI, which only needs 3 batches in expectation to find the best arm with probability $1 - \delta$ and achieves the asymptotic optimal sample complexity. We further proposed Opt-BBAI and theoretically showed that Opt-BBAI has a near-optimal non-asymptotic sample and batch complexity while still maintaining the asymptotic optimality as Tri-BBAI does.

In our experiments, although Tri-BBAI utilizes a limited number of batches, its sample complexity does not match that of Garivier and Kaufmann [17]. Designing a batched algorithm with sample complexity comparable to Garivier and Kaufmann [17], while maintaining a constant number of batches, presents an intriguing challenge.

As for future work, an interesting direction is to investigate whether our "checking for best arm elimination" could be beneficial to other elimination-based algorithms. Additionally, some research [3, 23] implied a strong correlation between batch complexity in batched bandit, and the memory complexity and pass complexity in streaming bandit. Thus, it could be valuable to assess if our techniques could enhance the results in the field of streaming bandit.

## Acknowledgements

We would like to thank the anonymous reviewers for their helpful comments. This research is funded by a Singapore Ministry of Education AcRF Tier 2 grant (A-8000423-00-00), by the National Research Foundation, Singapore under its AI Singapore Program (AISG Award No: AISG-PhD/2021-01004[T]), by the Ministry of Education, Singapore, under its MOE AcRF TIER 3 Grant (MOE-MOET32022-0001), by National Key R&D Program of China under Grant No. 2023YFF0725100, by the National Natural Science Foundation of China (NSFC) under Grant No. 62402410 and U22B2060, by Guangdong Basic and Applied Basic Research Foundation under Grant No. 2023A1515110131, and by Guangzhou Municipal Science and Technology Bureau under Grant No. 2023A03J0667 and 2024A04J4454.

In particular, T. Jin was supported by a Singapore Ministry of Education AcRF Tier 2 grant (A-8000423-00-00) and by the National Research Foundation, Singapore under its AI Singapore Program (AISG Award No: AISG-PhD/2021-01004[T]). X. Xiao was supported by the Ministry of Education, Singapore, under its MOE AcRF TIER 3 Grant (MOE-MOET32022-0001). J. Tang was partially supported by National Key R&D Program of China under Grant No. 2023YFF0725100, by the National Natural Science Foundation of China (NSFC) under Grant No. 62402410 and U22B2060, by Guangdong Basic and Applied Basic Research Foundation under Grant No. 2023A1515110131, and by Guangzhou Municipal Science and Technology Bureau under Grant No. 2023A03J0667 and 2024A04J4454. P. Xu was supported in part by the Whitehead Scholars Program at the Duke University School of Medicine. The views and conclusions in this paper are those of the authors and should not be interpreted as representing any funding agencies.

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

## Notations.

Denote by $\hat{\mu}_i^s$ the empirical mean of arm $i$ after its $s$-th pull and by $\hat{\mu}_i(t)$ the empirical mean of arm $i$ at time $t$ (i.e., after a total number of $t$ pulls are examined for all arms). We use $\Delta_i = \mu_1 - \mu_i$ for the gap between the optimal arm and the $i$-th arm. Throughout the paper, we use the following asymptotic notation. Specifically we use $f(\delta) \lesssim g(\delta)$ to denote that there exists some $\delta_0$, such that for any $\delta < \delta_0$, $f(\delta) \leq g(\delta)$. Similarly, we use $f(\delta) \gtrsim g(\delta)$ to denote that there exists some $\delta_0$, such that for any $\delta < \delta_0$, $f(\delta) \geq g(\delta)$.

## A  Computing $w^*(\mu)$ and $T^*(\mu)$

In this section, we introduce two useful lemmas that help us to calculate $\boldsymbol{w}^*(\boldsymbol{b})$ and $T^*(\boldsymbol{b})$ in Algorithm 1 and Algorithm 2, which are also used in [17]. For ease of presentation, we first define some useful notations. For every $c \in [0, 1]$, we define

$$I_c(\mu, \mu') := cd(\mu, c\mu + (1-c)\mu') + (1-c)d(\mu', c\mu + (1-c)\mu').$$

For any arm $i \in [n] \setminus \{1\}$, we define

$$g_i(x) := (1+x)I_{\frac{1}{1+x}}(\mu_1, \mu_i).$$

**Lemma A.1** (Garivier and Kaufmann 17, Lemma 3). For every $\boldsymbol{w} \in \mathcal{P}_n$,

$$\inf_{\boldsymbol{\lambda} \in \mathrm{Alt}(\boldsymbol{\mu})} \left( \sum_{i=1}^n w_i d(\mu_i, \lambda_i) \right) = \min_{i \neq 1} (w_1 + w_i) I_{\frac{w_1}{w_1 + w_i}}(\mu_1, \mu_i).$$

Besides,

$$T^*(\boldsymbol{\mu})^{-1} = \sup_{\boldsymbol{w} \in \mathcal{P}_n} \min_{i \neq 1} (w_1 + w_i) I_{\frac{w_1}{w_1 + w_i}}(\mu_1, \mu_i), \tag{A.1}$$

and

$$\boldsymbol{w}^*(\boldsymbol{\mu}) = \arg\max_{\boldsymbol{w} \in \mathcal{P}_n} \min_{i \neq 1} (w_1 + w_i) I_{\frac{w_1}{w_1 + w_i}}(\mu_1, \mu_i). \tag{A.2}$$

**Lemma A.2** (Garivier and Kaufmann 17, Theorem 5). For $i \in [n]$, let

$$w_i^*(\boldsymbol{\mu}) = \frac{x_i(y^*)}{\sum_{i \in [n]} x_i(y^*)},$$

where $y^*$ is the unique solution of the equation $F_{\boldsymbol{\mu}}(y) = 1$, and where

$$F_{\boldsymbol{\mu}} : y \mapsto \sum_{i=2}^n \frac{d\left(\mu_1, \frac{\mu_1 + x_i(y)\mu_i}{1 + x_i(y)}\right)}{d\left(\mu_i, \frac{\mu_1 + x_i(y)\mu_i}{1 + x_i(y)}\right)}.$$

in a continuous, increasing function on $[0, d(\mu_1, \mu_2))$ such that $F_{\boldsymbol{\mu}}(y) \to \infty$ when $y \to d(\mu_1, \mu_2)$.

## B  Proof of Theorems in Section 3

### B.1  Proof of Theorem 3.2

*Proof of Theorem 3.2.* The proof of Theorem 3.2 requires the following two Lemmas, which guarantees the error returned by Line 22 and Line 26 of Algorithm 1 respectively.

**Lemma B.1.** [17, Proposition 12] Let $\boldsymbol{\mu}$ be the exponential bandit model. Let $\delta \in (0, 1)$. For sufficiently small $\delta > 0$, the Chernoff's stopping rule Line 21 of Algorithm 1 with the threshold

$$\beta(t, \delta/2) = \log\left(\frac{2\log(2/\delta)t^\alpha}{\delta}\right),$$

ensures that $\mathbb{P}_{\boldsymbol{\mu}}(i^*(N_\delta) \neq 1) \leq \delta/2$.

Recall $\hat{\mu}_i^s$ is the empirical mean of arm $i$ after its $s$-th pull. Let $\mathcal{E}_1 = \{\forall s \geq L_3 \text{ and } i \in [n] : \hat{\mu}_i^s \in [\mu_i - \epsilon, \mu_i + \epsilon]\}$. The following lemma shows that $\mathbb{P}(\mathcal{E}_1^c) \leq \delta/2$, where $\mathcal{E}^c$ denotes the complement event of $\mathcal{E}$.

**Lemma B.2.** For sufficiently small $\delta > 0$[6], we have
$$\mathbb{P}(\mathcal{E}_1^c) \leq \delta/2.$$

If $\mathcal{E}_1$ happens, we have that at Line 26 of Algorithm 1, $\hat{\mu}_1(t) \geq \mu_1 - \epsilon \gtrsim \mu_i + \epsilon \geq \hat{\mu}_i(t)$ for all $i > 1$, which means $i^*(t) = 1$. Combining Lemma B.1 and Lemma B.2, Theorem 3.1 follows immediately. $\qquad\square$

## B.2 Proof of Theorem 3.1

*Proof of Theorem 3.1.* Recall in Algorithm 1, $b_1^q = \hat{\mu}_{i^*(t)}(t) - \epsilon$ and $b_i^q = \hat{\mu}_i(t) + \epsilon$ for all $i \in [n] \setminus \{i^*(t)\}$, where $t = \tau_q$. Let $\mathcal{E}_0$ be the intersection of the following four events:

$$\mathcal{E}_0^1 = \cap_{q\geq 1}\{b_1^q \in [\mu_1 - 3\epsilon/2, \mu_1 - \epsilon/2]\}, \tag{B.1a}$$

$$\mathcal{E}_0^2 = \cap_{q\geq 1}\{\text{for all } i \in [n] \setminus \{1\}, b_i^q \in [\mu_i + \epsilon/2, \mu_i + 3\epsilon/2]\}, \tag{B.1b}$$

$$\mathcal{E}_0^3 = \cap_{q\geq 0}\{\hat{\mu}_1(\tau_q) \geq \mu_1 - \epsilon/2\}, \tag{B.1c}$$

$$\mathcal{E}_0^4 = \cap_{q\geq 0}\{\text{for all } i \in [n] \setminus \{1\}, \hat{\mu}_i(\tau_q) \leq \mu_i + \epsilon/2\}. \tag{B.1d}$$

Here (B.1a) and (B.1b) ensure that the estimation of $\boldsymbol{w}^*(\boldsymbol{b}^q)$ and $T^*(\boldsymbol{b}^q)$ is close to $\boldsymbol{w}^*(\boldsymbol{\mu})$ and $T^*(\boldsymbol{\mu})$, and (B.1c) and (B.1d) ensure that arm 1 has the largest average reward at time $\tau_q$.

**Lemma B.3.** For sufficiently small $\delta > 0$, we have
$$\mathbb{P}((\mathcal{E}_0^c) \leq \frac{1}{(\log \delta^{-1})^2}.$$

**Lemma B.4.** If $\mathcal{E}_0$ holds, then for sufficiently small $\delta > 0$, we have

1. $T_i = \alpha w_i^*(\boldsymbol{\mu})T^*(\boldsymbol{\mu})\log(1/\delta) + o(\log(1/\delta))$;

2. $|w_i^*(\boldsymbol{b}^2) - w_i^*(\boldsymbol{b}^1)| \leq 1/\sqrt{n}$ for all $i \in [n]$;

3. $\min_{j\notin[n]\setminus\{i^*(\tau)\}} Z_j(\tau) \geq \beta(\tau, \delta/2)$.

From Lemma B.4, if $\delta$ is sufficiently small and $\mathcal{E}_0$ holds, then Algorithm 1 will return at Line 22. Besides, we note that the total number of pulls of any arm is no more than $L_3 = (\log(\delta^{-1}))^2$ times. Therefore,

$$\begin{aligned}
\mathbb{E}[N_\delta] &\leq nL_1 + \mathbb{1}\{\mathcal{E}_0\} \cdot (\alpha T^*(\boldsymbol{\mu})\log\delta^{-1} + o(n\log(1/\delta))) + \mathbb{1}\{\mathcal{E}_0^c\} \cdot (L_2 + nL_3) \\
&\leq \mathbb{1}\{\mathcal{E}_0\} \cdot \alpha T^*(\boldsymbol{\mu})\log\delta^{-1} + o(n\log(1/\delta)) + \mathbb{1}\{\mathcal{E}_0^c\}nL_3 \\
&\lesssim \mathbb{1}\{\mathcal{E}_0\} \cdot \alpha T^*(\boldsymbol{\mu})\log\delta^{-1} + n \\
&\leq \alpha T^*(\boldsymbol{\mu})\log\delta^{-1} + n.
\end{aligned} \tag{B.2}$$

We further have
$$\lim_{\delta\to 0}\frac{\mathbb{E}[N_\delta]}{\log\delta^{-1}} \leq \alpha T^*(\boldsymbol{\mu}).$$

This completes the proof. $\qquad\square$

## B.3 Proof of Theorem 3.3

*Proof of Theorem 3.3.* Stage I costs one batch. For the Stage II, note that $\mathbb{P}(\mathcal{E}_0^c) \leq \frac{1}{\log(\delta^{-1})^2}$ and if $\mathcal{E}_0$ occurs, then $|w_i^*(\boldsymbol{b}^2) - w_i^*(\boldsymbol{b}^1)| \leq 1/\sqrt{n}$ for all $i \in [n]$ (from the second statement of Lemma B.4), which means Stage II costs 2 batches. Moreover, from Lemma B.4, if $\delta$ is sufficiently small and $\mathcal{E}_0$ holds, then Algorithm 1 will return at Line 22. Otherwise, the algorithm goes to Stage IV and costs one batch. Therefore, for sufficiently small $\delta$, the expected number of batches
$$\leq 1 + \mathbb{P}(\mathcal{E}_0^c) \cdot \log(1/\delta) + 2 + \mathbb{P}(\mathcal{E}_0^c) = 3 + o(1).$$

Finally, if $\mathcal{E}_0$ is true, the algorithm costs 3 batches. Therefore, with probability $1 - 1/\log(1/\delta^2)$, Algorithm 1 costs 3 batches. $\qquad\square$

---
[6]If event $E$ occur for a sufficiently small $\delta > 0$, it signifies that there exists a $\delta_0 \in (0, 1)$ such that for all $\delta \leq \delta_0$, event $E$ consistently holds.

## B.4  Proof of Supporting Lemmas

The proof of Lemma B.3 requires the following useful inequalities.

**Lemma B.5** (Maximal Inequality). Let $N$ and $M$ be two positive integers, let $\gamma > 0$, and $\hat{\mu}_n$ be the empirical mean of $n$ random variables i.i.d. according to the arm's distribution with mean $\mu$. Then, for $x \leq \mu$,

$$\mathbb{P}(\exists N \leq n \leq M, \hat{\mu}_n \leq x) \leq e^{-N(x-\mu)^2/(2V_0)}, \tag{B.3}$$

and for every $x \geq \mu$,

$$\mathbb{P}(\exists N \leq n \leq M, \hat{\mu}_n \geq x) \leq e^{-N(x-\mu)^2/(2V_1)}, \tag{B.4}$$

where $V_0$ is the maximum variance of arm's distribution with mean $\mu \in [x, \mu]$ and $V_1$ is the maximum variance of arm's distribution with mean $\mu \in [\mu, x]$.

Note that Lemma B.5 is an improved version of Lemma 4 of [36], where we use a much smaller variance upper bound to tighten the inequalities for the case $x \leq \mu$ and the case $x \geq \mu$ respectively.

**Lemma B.6.** [24, Proposition 1] For $\epsilon > 0$ and $\mu \leq \mu' - \epsilon$,

$$d(\mu, \mu') \geq d(\mu, \mu' - \epsilon), \quad \text{and} \quad d(\mu, \mu') \leq d(\mu - \epsilon, \mu'). \tag{B.5}$$

Now, we are ready to prove Lemma B.3.

*Proof of Lemma B.3.* Let $V$ be the maximum variance of reward distribution with mean $\mu \in [\mu_n, \mu_1]$. From Lemma B.5, we have that for $s \geq L_1$,

$$\begin{aligned}
&\mathbb{P}(\exists s \geq L_1 : \hat{\mu}_1^s \notin [\mu_1 - \epsilon/2, \mu_1 + \epsilon/2]) \\
&\leq \mathbb{P}(\exists s \geq L_1 : \hat{\mu}_1^s \geq \mu_1 + \epsilon/2) + \mathbb{P}(\exists s \geq L_1 : \hat{\mu}_1^s \leq \mu_1 - \epsilon/2) \\
&\leq 2e^{-L_1(\epsilon/2)^2/(2V)} \\
&\lesssim \frac{1}{n(\log \delta^{-1})^2},
\end{aligned} \tag{B.6}$$

where the second inequality is from Lemma B.5, and the last inequality is due to that for sufficiently small $\delta > 0$, $L_1 = \sqrt{\log \delta^{-1}} \geq 8V/\epsilon^2 \log(n(\log \delta^{-1})^2)$. Similarly, for $i \in [n] \setminus \{1\}$,

$$\mathbb{P}(\exists s \geq L_1 : \hat{\mu}_i^s \notin [\mu_i - \epsilon/2, \mu_i + \epsilon/2]) \lesssim \frac{1}{n(\log \delta^{-1})^2}. \tag{B.7}$$

Define events:

$$\mathcal{A}_1 = \{\forall s \geq L_1 : \hat{\mu}_1^s \in [\mu_1 - \epsilon/2, \mu_1 + \epsilon/2]\}$$

and

$$\mathcal{A}_2 = \{\text{for all } i \in [n] \setminus \{1\}, \forall s \geq L_1 : \hat{\mu}_i^s \in [\mu_i - \epsilon/2, \mu_i + \epsilon/2]\}.$$

Assume that both events $\mathcal{A}_1$ and $\mathcal{A}_2$ hold. Then, we have

$$\hat{\mu}_1^{L_1} \geq \mu_1 - \epsilon/2 \geq \mu_i + \epsilon/2 + \Delta_i - \epsilon \geq \hat{\mu}_i^{L_1},$$

where in the last inequality we assumed $\epsilon \leq \min_{i \neq 1} \Delta_i$. This further implies $i^*(\tau_q) = 1$ for any $q \geq 0$ and thus $b_1^{q-1} = \hat{\mu}_1(\tau_q) - \epsilon$. Therefore, we have for any $q \geq 1$

1. $b_1^q = \hat{\mu}_1(\tau_{q-1}) - \epsilon \in [\mu_1 - 3\epsilon/2, \mu_1 - \epsilon/2]$;

2. $\forall i \in [n] \setminus \{1\}, b_i^q = \hat{\mu}_i(\tau_{q-1}) + \epsilon \in [\mu_i - \epsilon/2, \mu_1 + \epsilon/2]$;

3. $\hat{\mu}_1(\tau_{q-1}) \geq \mu_1 - \epsilon/2$;

4. and $\forall i \in [n] \setminus \{1\}, \hat{\mu}_i(\tau_{q-1}) \leq \mu_i + \epsilon/2$,

which means $\mathcal{E}_0$ defined in (B.1) occurs. Therefore, we have

$$
\begin{aligned}
\mathbb{P}(\mathcal{E}_0) &\geq \mathbb{P}(\mathcal{A}_1 \cap \mathcal{A}_2) \\
&= \mathbb{P}\Bigg( \{\forall s \geq L_1 : \hat{\mu}_1^s \in [\mu_1 - \epsilon/2, \mu_1 + \epsilon/2]\} \bigcap_{i:i\in[n]\setminus\{1\}} \{\forall s \geq L_1 : \hat{\mu}_i^s \in [\mu_i - \epsilon/2, \mu_i + \epsilon/2]\}\Bigg) \\
&= \mathbb{P}\Bigg( \{\exists s \geq L_1 : \hat{\mu}_1^s \notin [\mu_1 - \epsilon/2, \mu_1 + \epsilon/2]\}^c \bigcap_{i:i\in[n]\setminus\{1\}} \{\exists s \geq L_1 : \hat{\mu}_i^s \notin [\mu_i - \epsilon/2, \mu_i + \epsilon/2]\}^c\Bigg) \\
&\geq 1 - \mathbb{P}\left( \exists s \geq L_1 : \hat{\mu}_1^s \notin \left[\mu_1 - \frac{\epsilon}{2}, \mu_1 + \frac{\epsilon}{2}\right]\right) \\
&\quad - \sum_{i:i\in[n]\setminus\{1\}} \mathbb{P}\left( \exists s \geq L_1 : \hat{\mu}_i^s \notin \left[\mu_i - \frac{\epsilon}{2}, \mu_i + \frac{\epsilon}{2}\right]\right) \\
&\geq 1 - (\log \delta^{-1})^{-2},
\end{aligned}
$$

where the last inequality is due to (B.6) and (B.7). $\square$

*Proof of Lemma B.4.* Assume $\mathcal{E}_0$ is true. Recall that $\Delta_2 = \min_{i\in[n]\setminus\{1\}} \mu_1 - \mu_i$. For sufficiently small $\delta > 0$ such that $\epsilon < \Delta_2/8$, we have $b_1^q \in [\mu_1 - \Delta_2/4, \mu_1]$ and $b_i^q \in [\mu_i, \mu_i + \Delta_2/4]$. Let $\mathcal{L}(\boldsymbol{b}) = \{\boldsymbol{\lambda} \in S : \lambda_1 \in [\mu_1 - \Delta_2/4, \mu_1] \text{ and for any } i \in [n] \setminus \{1\}, \lambda_i \in [\mu_i, \mu_i + \Delta_2/4]\}$. Therefore

$$
\alpha w_i^*(\boldsymbol{b}^q) T^*(\boldsymbol{b}^q) \leq \max_{\boldsymbol{b}'\in\mathcal{L}(\boldsymbol{b})} \alpha w_i^*(\boldsymbol{b}') T^*(\boldsymbol{b}') < \infty.
$$

Since $\max_{\boldsymbol{b}'\in\mathcal{L}(\boldsymbol{b})} \alpha w_i^*(\boldsymbol{b}') T^*(\boldsymbol{b}') < \infty$ is independent of $\delta$, we have

$$
\alpha w_i^*(\boldsymbol{b}^q) T^*(\boldsymbol{b}^q) \leq \max_{\boldsymbol{b}'\in\mathcal{L}(\boldsymbol{b})} \alpha w_i^*(\boldsymbol{b}') T^*(\boldsymbol{b}') \lesssim \log \log \delta^{-1}/n = L_2,
$$

which implies $T_i^q = \min\{\alpha w_i^*(\boldsymbol{b}^q) T^*(\boldsymbol{b}^q) \log \delta^{-1}, L_2\} = \alpha w_i^*(\boldsymbol{b}^q) T^*(\boldsymbol{b}^q) \log \delta^{-1}$. Besides, as $w^*(\boldsymbol{b}^q)$ and $T^*(\boldsymbol{b}^q)$ are continuous on $\boldsymbol{b}^q$, we have that as $\boldsymbol{b}^q \to \boldsymbol{\mu}$, $w^*(\boldsymbol{b}^q) \to w^*(\boldsymbol{\mu})$ and $T^*(\boldsymbol{b}^q) \to T^*(\boldsymbol{\mu})$. Note that $|b_i^q - \mu_i| \leq 2\epsilon$ and $\epsilon$ approaches 0 as $\delta$ approaches 0. Recall $\tau$ is the stopping time of Stage II and $T_i$ is the number of pulls of arm $i$ at time $\tau$. Therefore, for sufficiently small $\delta$,

$$
T_i := \max_{p:p\in\mathbb{N}, p\geq 1} T_i^p = \alpha w_i^*(\boldsymbol{\mu}) T^*(\boldsymbol{\mu}) \log \delta^{-1} + o(\log(1/\delta)).
$$

Moreover, for sufficient smaller $\delta$ and $\forall i \in [n]$, $|w_i^*(\boldsymbol{b}^q) - w_i^*(\boldsymbol{\mu})| = o(1)$ and thus

$$
|w_i^*(\boldsymbol{b}^1) - w_i^*(\boldsymbol{b}^2)| < 1/\sqrt{n},
$$

which means the condition in Line 15 is satisfied and the Algorithm goes to Stage III.

Assume $\tau = \tau_{q'}$. We have

$$
\begin{aligned}
Z_{1i}(\tau) &= N_1(\tau) \cdot d(\hat{\mu}_1(\tau), \hat{\mu}_{1i}(\tau)) + N_i(\tau) \cdot d(\hat{\mu}_i(\tau), \hat{\mu}_{1i}(\tau)) \\
&= N_1(\tau_{q'}) \cdot d(\hat{\mu}_1(\tau), \hat{\mu}_{1i}(\tau)) + N_i(\tau_{q'}) \cdot d(\hat{\mu}_i(\tau), \hat{\mu}_{1i}(\tau)) \\
&\geq \alpha \log \delta^{-1} \cdot \left( \frac{w_1^*(\boldsymbol{b}^{q'}) d(\hat{\mu}_1(\tau), \hat{\mu}_{1i}(\tau))}{T^*(\boldsymbol{b}^{q'})^{-1}} + \frac{w_i^*(\boldsymbol{b}^{q'}) d(\hat{\mu}_1(\tau), \hat{\mu}_{1i}(\tau))}{T^*(\boldsymbol{b}^{q'})^{-1}} \right),
\end{aligned} \tag{B.8}
$$

where the last inequality is due to $T_i \geq T_i^{q'} = \alpha w_i^*(\boldsymbol{b}^{q'}) T^*(\boldsymbol{b}^{q'}) \log(1/\delta)$. In what follows, we will show

$$
\frac{w_1^*(\boldsymbol{b}^{q'}) \cdot d(\hat{\mu}_1(\tau), \hat{\mu}_{1i}(\tau))}{T^*(\boldsymbol{b}^{q'})^{-1}} + \frac{w_i^*(\boldsymbol{b}^{q'}) \cdot d(\hat{\mu}_i(\tau), \hat{\mu}_{1i}(\tau))}{T^*(\boldsymbol{b}^{q'})^{-1}} \geq 1.
$$

The following minimization problem is a convex optimization problem

$$
\min_{\lambda_{1w}:\lambda_{1w}\leq\lambda_{iw}} w_1^*(\boldsymbol{b}^{q'}) \cdot d(\hat{\mu}_1(\tau), \lambda_{1w}) + w_i^*(\boldsymbol{b}^{q'}) \cdot d(\hat{\mu}_i(\tau), \lambda_{iw}),
$$

which is solved when we have $\lambda_{1w} = \lambda_{iw} = \hat{\mu}_{1i}(\tau)$. The proof of this lemma is based on the assumption that $\mathcal{E}_0$ occurs. Note that $\mathcal{E}_0$ occurs, then $b_1^{q'} = \hat{\mu}_1(\tau_{q'-1}) - \epsilon \leq \mu_1 - \epsilon/2 \leq \hat{\mu}_1(\tau)$ and $b_i^{q'} = \hat{\mu}_1(\tau_{q'-1}) + \epsilon > \hat{\mu}_i(\tau)$ for all $i \in [n] \setminus \{1\}$. Therefore, $\hat{\mu}_1(\tau) \geq b_1^{q'} \gtrsim b_i^{q'} \geq \hat{\mu}_i(\tau)$. Let

$$\lambda_{w^*(\boldsymbol{b}^{q'})}(b_1^{q'}, b_i^{q'}) = \frac{w_1^*(\boldsymbol{b}^{q'})}{w_1^*(\boldsymbol{b}^{q'}) + w_i^*(\boldsymbol{b}^{q'})} \cdot b_1^q + \frac{w_i^*(\boldsymbol{b}^{q'})}{w_1^*(\boldsymbol{b}^{q'}) + w_i^*(\boldsymbol{b}^{q'})} \cdot b_i^q.$$

Then, $\lambda_{w^*(\boldsymbol{b})}(b_1^{q'}, b_i^{q'})$ is the solution to minimizing $w_1^*(\boldsymbol{b}^{q'}) \cdot d(b_1^{q'}, x) + w_i^*(\boldsymbol{b}^{q'}) \cdot d(b_i^{q'}, x)$ for $x \in (b_i^{q'}, b_1^{q'})$.

**Case 1:** if $\hat{\mu}_{1i}(\tau) \in (b_1^{q'}, \hat{\mu}_1(\tau))$, then

$$w_1^*(\boldsymbol{b}^{q'}) \cdot d(\hat{\mu}_1(\tau), \hat{\mu}_{1i}(\tau)) + w_i^*(\boldsymbol{b}^{q'}) \cdot d(\hat{\mu}_i(\tau), \hat{\mu}_{1i}(\tau))$$
$$\geq w_1^*(\boldsymbol{b}^{q'}) \cdot d(\hat{\mu}_1(\tau), \hat{\mu}_1(\tau)) + w_i^*(\boldsymbol{b}^{q'}) \cdot d(\hat{\mu}_i(\tau), b_1^{q'})$$
$$\geq w_1^*(\boldsymbol{b}^{q'}) \cdot d(b_1^{q'}, b_1^{q'}) + w_i^*(\boldsymbol{b}^{q'}) \cdot d(b_i^{q'}, b_1^{q'})$$
$$\geq w_1^*(\boldsymbol{b}^{q'}) \cdot d(b_1^{q'}, \lambda_{w^*(\boldsymbol{b}^{q'})}(b_1^{q'}, b_i^{q'})) + w_i^*(\boldsymbol{b}^{q'}) \cdot d(b_i^{q'}, \lambda_{w^*(\boldsymbol{b}^{q'})}(b_1^{q'}, b_i^{q'})), \qquad \text{(B.9)}$$

where the first and second inequalities are due to (B.5) and the last inequality is due to the fact that $w_1^*(\boldsymbol{b}^{q'}) \cdot d(b_1^{q'}, x) + w_i^*(\boldsymbol{b}^{q'}) \cdot d(b_i^{q'}, x)$ achieves it minimum at $x = \lambda_{w^*(\boldsymbol{b}^{q'})}(b_1^{q'}, b_i^{q'})$.

**Case 2:** if $\hat{\mu}_{1i}(\tau) \in (b_i^{q'}, b_1^{q'})$, we have

$$w_1^*(\boldsymbol{b}^{q'}) \cdot d(\hat{\mu}_1(\tau), \hat{\mu}_{1i}(\tau)) + w_i^*(\boldsymbol{b}^{q'}) \cdot d(\hat{\mu}_i(\tau), \hat{\mu}_{1i}(\tau))$$
$$\geq w_1^*(\boldsymbol{b}^{q'}) \cdot d(b_1^{q'}, \hat{\mu}_{1i}(\tau)) + w_i^*(\boldsymbol{b}^{q'}) \cdot d(b_i^{q'}, \hat{\mu}_{1i}(\tau))$$
$$\geq w_1^*(\boldsymbol{b}^{q'}) \cdot d(b_1^{q'}, \lambda_{w^*(\boldsymbol{b}^{q'})}(b_1^{q'}, b_i^{q'})) + w_i^*(\boldsymbol{b}^{q'}) \cdot d(b_i^{q'}, \lambda_{w^*(\boldsymbol{b}^{q'})}(b_1^{q'}, b_i^{q'})), \qquad \text{(B.10)}$$

where the first inequality is due to (B.5) and the last inequality is due to the fact that for $x \in (b_i^{q'}, b_1^{q'})$, $w_1^*(\boldsymbol{b}^{q'}) \cdot d(b_1^{q'}, x) + w_i^*(\boldsymbol{b}^{q'}) \cdot d(b_i^{q'}, x)$ achieves its minimum at $x = \lambda_{w^*(\boldsymbol{b}^{q'})}(b_1 \ldots \ldots q', b_i^{q'})$.

**Case 3:** if $\hat{\mu}_{1i}(\tau) \in (\hat{\mu}_i(\tau), b_i^{q'})$, similar to (B.9) and (B.10), we have

$$w_1^*(\boldsymbol{b}^{q'}) \cdot d(\hat{\mu}_1(\tau), \hat{\mu}_{1i}(\tau)) + w_i^*(\boldsymbol{b}^{q'}) \cdot d(\hat{\mu}_i(\tau), \hat{\mu}_{1i}(\tau))$$
$$\geq w_1^*(\boldsymbol{b}^{q'}) \cdot d(b_1^{q'}, \lambda_{w^*(\boldsymbol{b}^{q'})}(b_1^{q'}, b_i^{q'})) + w_i^*(\boldsymbol{b}^{q'}) \cdot d(b_i^{q'}, \lambda_{w^*(\boldsymbol{b}^{q'})}(b_1^{q'}, b_i^{q'})). \qquad \text{(B.11)}$$

Combine all three cases, we obtain

$$w_1^*(\boldsymbol{b}^{q'}) \cdot d(\hat{\mu}_1(\tau), \hat{\mu}_{1i}(\tau)) + w_i^*(\boldsymbol{b}^{q'}) \cdot d(\hat{\mu}_i(\tau), \hat{\mu}_{1i}(\tau))$$
$$\geq w_1^*(\boldsymbol{b}^{q'}) \cdot d(b_1, \lambda_{w^*(\boldsymbol{b}^{q'})}(b_1^{q'}, b_i^{q'})) + w_i^*(\boldsymbol{b}^{q'}) \cdot d(b_i^{q'}, \lambda_{w^*(\boldsymbol{b}^{q'})}(b_1^{q'}, b_i^{q'})). \qquad \text{(B.12)}$$

Note that $\hat{\mu}_{1i}(\tau) = \lambda_{w^*(\boldsymbol{b}^{q'})}(\hat{\mu}_1(\tau), \hat{\mu}_i(\tau))$, we have

$$\frac{w_1^*(\boldsymbol{b}^{q'}) \cdot d(\hat{\mu}_1(\tau), \hat{\mu}_{1i}(\tau))}{T^*(\boldsymbol{b}^{q'})^{-1}} + \frac{w_i^*(\boldsymbol{b}^{q'}) \cdot d(\hat{\mu}_1(\tau), \hat{\mu}_{1i}(\tau))}{T^*(\boldsymbol{b}^{q'})^{-1}}$$
$$\geq \frac{w_1^*(\boldsymbol{b}^{q'}) \cdot d(b_1^{q'}, \lambda_{w^*(\boldsymbol{b}^{q'})}(b_1^{q'}, b_i^{q'}))}{T^*(\boldsymbol{b}^{q'})^{-1}} + \frac{w_i^*(\boldsymbol{b}^{q'}) \cdot d(b_i^{q'}, \lambda_{w^*(\boldsymbol{b}^{q'})}(b_1^{q'}, b_i^{q'}))}{T^*(\boldsymbol{b}^{q'})^{-1}}$$
$$= 1, \qquad \text{(B.13)}$$

where the first inequality is due to (B.12) and the last inequality is due to (A.1). Note that conditioned on event $\mathcal{E}_0$, for sufficiently small $\delta > 0$, we have $1 = i^*(\tau)$. Therefore, for any $i \in [n] \setminus \{1\}$,

$$Z_i(\tau) = Z_{1i}(\tau)$$
$$= \alpha \log \delta^{-1} \cdot \left( \frac{w_1^*(\boldsymbol{b}^{q'}) \cdot d(\hat{\mu}_1(\tau), \hat{\mu}_{1i}(\tau))}{T^*(\boldsymbol{b}^{q'})^{-1}} + \frac{w_i^*(\boldsymbol{b}^{q'}) \cdot d(\hat{\mu}_1(\tau), \hat{\mu}_{1i}(\tau))}{T^*(\boldsymbol{b}^{q'})^{-1}} \right)$$
$$\geq \alpha \log \delta^{-1},$$

where the first equality is from (3.3), the second equality is from (B.8), and the last inequality is from (B.13). Finally, for $\alpha \geq 1 + 6 \log \log \delta^{-1} / \log \delta^{-1}$, we obtain

$$Z_i(\tau) \geq \log\left(\frac{(\log \delta^{-1})^6}{\delta}\right) \gtrsim \log\left(\frac{2(\log(2/\delta))) \cdot (nL_2 + nL_1)^\alpha}{\delta}\right) = \beta(\tau, \delta/2),$$

which completes the proof. $\qquad\square$

### B.5 Proof of Maximum Inequality

*Proof of Lemma B.5.* Recall that the result of Lemma 4 of Ménard and Garivier [36] is: for $\hat{\mu}_n \leq \mu$,

$$\mathbb{P}(\exists N \leq n \leq M, d(\hat{\mu}_n, \mu) \geq \gamma) \leq e^{-N\gamma}. \tag{B.14}$$

By Lemma 1 of Harremoës [20], we have

$$d(x, \mu) = \int_x^\mu \frac{y - x}{V(y)} \mathrm{d}y \leq \frac{(x - \mu)^2}{2V_0}, \tag{B.15}$$

where $V(y)$ is the variance of the distribution with mean $y$. As a simple consequence of (B.15) and (B.14), we have

$$\mathbb{P}(\exists N \leq n \leq M, \hat{\mu}_n \leq x) \leq e^{-N(x-\mu)^2/(2V_0)}. \tag{B.16}$$

For the case when $\hat{\mu}_n \geq \mu$, we can directly follow the idea of Lemma 4 of Ménard and Garivier [36]. For the case $\hat{\mu}_n \geq \mu$, we aim to show

$$\mathbb{P}(\exists N \leq n \leq M, d(\hat{\mu}_n, \mu) \geq \gamma) \leq e^{-N\gamma}. \tag{B.17}$$

We let $M(\lambda)$ be the log-moment generating function of $\nu_\theta$. Recall that $\frac{\mathrm{d}\nu_\theta}{\mathrm{d}\rho}(x) = \exp(x\theta - b(\theta))$. We use the following properties of one exponential family.

1. $M(\lambda) = b(\theta + \lambda) - b(\theta)$;

2. $\mathrm{KL}(\nu_{\theta_1}, \nu_\theta) = b(\theta) - b(\theta_1) + b'(\theta_1)(\theta - \theta_1)$;

3. $\mathbb{E}[\nu_\theta] = b'(\theta)$.

Let $\mathbb{E}[\nu_\theta] = \mu$. Let $\lambda = \theta_1 - \theta$, $z = b'(\theta_1) > \mu$, and $\gamma = d(z, \mu)$, we have

$$\gamma = d(z, \mu) = \mathrm{KL}(\nu_{\theta_1}, \nu_\theta) = \lambda z - M(\lambda).$$

Since $d(z, \mu)$ is monotone increasing for $z > \mu$, we obtain $\lambda > 0$. If event $\{\exists N \leq n \leq M, d(\hat{\mu}_n, \mu) \geq \gamma\}$ and $\hat{\mu}_n \geq \mu$, one have

$$\hat{\mu}_n \geq \mu, \qquad \lambda\hat{\mu}_n - M(\lambda) \geq \lambda z - M(\lambda) = \gamma, \qquad \lambda n\hat{\mu}_n - nM(\lambda) \geq N\gamma.$$

By Doob's maximal inequality for the exponential martingale $\exp(\lambda n\hat{\mu}_n - nM(\lambda))$,

$$\mathbb{P}(\exists N \leq n \leq M, d(\hat{\mu}_n, \mu) \geq \gamma) \leq \mathbb{P}(\exists N \leq n \leq M, \lambda n\hat{\mu}_n - nM(\lambda) \geq N\gamma)$$
$$\leq e^{-N\gamma}. \tag{B.18}$$

Combining above inequality and (B.15), we obtain for $\hat{\mu}_n \geq \mu$

$$\mathbb{P}(\exists N \leq n \leq M, d(\hat{\mu}_n, \mu) \geq \gamma) \leq e^{-N(x-\mu)^2/(2V_1)}. \tag{B.19}$$

This completes the proof. $\qquad\square$

## C  Proof of Theorems in Section 4

In this proof, we define "one round" as a single iteration of the **While Loop** of Algorithm 2.

*Proof of Theorem 4.2.* **Proof of Correctness.** We first show that the best arm is not eliminated by Line 11 for any round $r$. For any $l$, let $\mathcal{E}^l$ be defined as follows.

$$\mathcal{E}^l = \{1 \in S_r, \forall r \leq l\}. \tag{C.1}$$

**Lemma C.1.** Line 11 of Algorithm 2 satisfies

$$\sum_{r \geq 1} \mathbb{P}\left( \hat{p}_1^r \geq \hat{p}_*^r - \frac{\epsilon_r}{4} \;\middle|\; \mathcal{E}^r \right) \geq 1 - \frac{\delta}{8}.$$

To streamline our presentation, we define $\ell_{js} = s$, $\gamma_{js} = (\delta_j)^{2^s}$, and $\hat{p}_i^{js}$ as the $s$th updates of parameters $\ell_j$, $\gamma_j$, and $\hat{p}_i^j$, respectively, within the second **For Loop**. For any fixed $j$, we define $U_j(s)$ to be the following set of rounds, where $\ell_j$ remains to be the same value $s$:

$$U_j(s) := \{r = 1, 2, \ldots : \ell_j = s \text{ at round } r \text{ of Algorithm } 2\}. \tag{C.2}$$

Then the condition of Line 21 could be represented as

$$\exists i \in S_j \text{ and } r \in U_j(s), \; \hat{p}_i^{js} \geq \hat{p}_*^r - \frac{\epsilon_j}{2}.$$

The following lemma shows that with high probability, Algorithm 2 will not return at Line 22.

**Lemma C.2.** Line 21 of Algorithm 2 satisfies

$$\mathbb{P}\left( \bigcap_{j \geq 1} \bigcap_{s > 1} \bigcap_{r \in U_j(s)} \left\{ \max_{i \in S_j} \hat{p}_i^{js} < \hat{p}_*^r - \frac{\epsilon_j}{2} \right\} \right) \geq 1 - \frac{3\delta}{16}.$$

Note that Algorithm 2 will return arm 1 at Line 24 if 1: Arm 1 is always maintained in $S_r$, and 2: Algorithm 2 never return at Line 22.

According to Lemma C.1, there is a probability of at least $1 - \delta/8$ that Arm 1 will never be eliminated at Line 11. Given that Arm 1 is never eliminated at Line 11, Lemma C.2 suggests that there is a probability of at least $1 - 3\delta/16$ that we will never loop back at Line 22. Hence, with a probability of $1 - \delta/2$, Stage IV will identify the optimal arm. By incorporating the results of Lemma B.1, we find that with a probability exceeding $1 - \delta/2 - \delta/2$, which is greater than $1 - \delta$, Algorithm 2 will return the optimal arm.

**Proof of Sample Complexity.** Let $N'$ be the sample complexity of Stage IV. Let $\mathcal{E} = \cap_{r > 1} \mathcal{E}^r$. The following lemma shows the sample complexity conditioned on event $\mathcal{E}$.

**Lemma C.3.** The expected sample complexity $N'$ conditioned on event $\mathcal{E}$ has the order

$$\mathbb{E}[N' \mid \mathcal{E}] = O\left( \sum_{i > 1} \frac{\log\left(n/\delta \log(\Delta_i^{-1})\right)}{\Delta_i^2} \right).$$

Now, we consider the expected sample complexity if the best arm is eliminated at some round $r$. We prove the following lemma.

**Lemma C.4.** The sum of expected sample complexity $N'$ at each loop has the order

$$\sum_{r \geq 1} \mathbb{E}[N' \mathbb{1}\{\mathcal{E}^r, (\mathcal{E}^{r+1})^c\}] = O\left( \sum_{i > 1} \frac{\log\left(n/\delta \log(\Delta_i^{-1})\right)}{\Delta_i^2} \right).$$

The number of pulls of all arms in the first three stages is no more than $\max\{n\sqrt{\log \delta^{-1}}, L_2\}$. By combining the above results, Lemma C.3 and Lemma C.4 together, we can obtain the following total sample complexity

$$\mathbb{E}[N_\delta] = \mathcal{O}\left( (\log \delta^{-1}) \cdot \log\log \delta^{-1} + \sum_{i > 1} \frac{\log\left(n\delta^{-1} \log \Delta_i^{-1}\right)}{\Delta_i^2} \right).$$

Since in the non-asymptotic setting, $\delta$ is finite. Then

$$\mathbb{E}[N_\delta] = \mathcal{O}\left( \sum_{i > 1} \frac{\log\left(n \log \Delta_i^{-1}\right)}{\Delta_i^2} \right).$$

**Proof of Batch Complexity.** Just as we discussed in the proof of sample complexity, we first demonstrate the batch complexity conditioned on the event of $\mathcal{E}$.

**Lemma C.5.** Conditional on event $\mathcal{E}$, Algorithm 2 conducts $\mathcal{O}(\log(1/\Delta_2^{-1}))$ batches in expectation.

Let $B'$ be the number of batches we used in Stage IV. We consider the expected batch complexity if the best arm is eliminated at some round $r$. We prove the following lemma.

**Lemma C.6.** The sum of expected batch complexity $B'$ at each loop has the order

$$\sum_{r \geq 1} \mathbb{E}[B' \mathbb{1}\{\mathcal{E}^r, (\mathcal{E}^{r+1})^c\}] = \mathcal{O}(\log(1/\Delta_2^{-1})).$$

By combining Lemma C.5, Lemma C.6, the fact that the first three stages use $O(\log(1/\delta))$ batches, and $\delta$ is a constant in non-asymptotic setting, the batch complexity of Algorithm 2 is $\mathcal{O}(\log(1/\Delta_2^{-1}))$.
$\square$

*Proof of Theorem 4.3.* Let $\mathcal{E}_0$ be the event defined in (B.1) in the proof of Theorem 3.1. From Lemma B.4, if $\delta$ is sufficiently small and $\mathcal{E}_0$ holds, then Algorithm 2 will return at Stage III. Note that the expected number of pulls in Stage IV is $\lesssim (\log \delta^{-1})^2$. Hence, the expected number of pulls of all arms is no more than $2(\log \delta^{-1})^2$ times. Therefore, for any $\epsilon' > 0$

$$\begin{aligned}
\mathbb{E}[N_\delta] &\lesssim \alpha T^*(\boldsymbol{b}) \log \delta^{-1} + o(\log(1/\delta)) + \mathbb{1}\{\mathcal{E}_0^c\} 2n(\log \delta^{-1})^2 \\
&\lesssim \alpha T^*(\boldsymbol{\mu}) \log \delta^{-1} + 2n.
\end{aligned} \tag{C.3}$$

Therefore,

$$\lim_{\delta \to 0} \frac{\mathbb{E}_{\boldsymbol{\mu}}[N_\delta]}{\log \delta^{-1}} \leq \alpha T^*(\boldsymbol{\mu}),$$

Moreover, there exists some universal constant $C$, such that the expected number of batches used in Stage IV is $\lesssim C \log(1/\Delta_2) \cdot \mathbb{1}\{\mathcal{E}_0^c\} \lesssim o(1)$, where the last inequality is due to Lemma B.3. Form Theorem 3.3, the batch complexity of the first three Stage is $3 + o(1)$. Therefore, the total expected batch complexity is $3 + o(1)$.
$\square$

# D   Proof of Supporting Lemmas

The proof of the supporting lemmas requires the following concentration inequalities.

**Lemma D.1.** [41, Theorem 2.2.6] Let $X_1, \ldots, X_k \in [0, 1]$ be independent bounded random variables with mean $\mu$. Then for any $\epsilon > 0$

$$\mathbb{P}(\hat{\mu} \geq \mu + \epsilon) \leq \exp\left(-\frac{k\epsilon^2}{2}\right) \quad \text{and} \tag{D.1}$$

$$\mathbb{P}(\hat{\mu} \leq \mu - \epsilon) \leq \exp\left(-\frac{k\epsilon^2}{2}\right), \tag{D.2}$$

where $\hat{\mu} = 1/k \sum_{t=1}^{k} X_t$.

## D.1   Proof of Lemma C.1

From Lemma D.1,

$$\mathbb{P}\left(|\hat{p}_i^r - \mu_i| \geq \frac{\epsilon_r}{8}\right) \leq 2 \exp\left(-\frac{2 d_r \epsilon_r^2}{64}\right) = \delta_r.$$

Applying union bound, we have

$$\begin{aligned}
\mathbb{P}\left(\bigcup_{r \geq 1} \bigcup_{i \in [n]} \left\{|\hat{p}_i^r - \mu_i| \geq \frac{\epsilon_r}{8}\right\}\right) &\leq \sum_{r > 1} \sum_{i \in [n]} \delta_r \\
&\leq \sum_{r > 1} \frac{\delta}{40\pi^2 \cdot r^2} \\
&\leq \frac{\delta}{8}.
\end{aligned} \tag{D.3}$$

The aforementioned inequality suggests that, with a probability of at least $1 - \delta/8$, the condition $|\hat{p}_i^r - \mu_i| \leq \frac{\epsilon_r}{8}$ holds true for any $i \in [n]$ and $r \geq 1$. Then, for any $r > 1$,

$$\hat{p}_1^r \geq \mu_1 - \frac{\epsilon_r}{8} \geq \mu_* - \frac{\epsilon_r}{8} \geq \hat{p}_*^r - \frac{\epsilon_r}{4}.$$

Therefore,

$$\sum_{r \geq 1} \mathbb{P}\left(\hat{p}_1^r \geq \hat{p}_*^r - \frac{\epsilon_r}{4} \,\Big|\, \mathcal{E}^r\right) \geq 1 - \frac{\delta}{8}.$$

## D.2    Proof of Lemma C.2

From Lemma D.1,

$$\mathbb{P}\left(|\hat{p}_i^{js} - \mu_i| \geq \frac{\epsilon_j}{8}\right) \leq \gamma_{js} = (\delta_j)^{2^s}.$$

Applying union bound, we have

$$\mathbb{P}\left(\bigcup_{j \geq 1} \bigcup_{s > 1} \bigcup_{i \in S_j} \left\{|\hat{p}_i^{js} - \mu_i| \geq \frac{\epsilon_j}{8}\right\}\right) \leq \sum_{j \geq 1} \sum_{s > 1} \sum_{i \in S_j} (\delta_j)^{2^s}$$

$$\leq \sum_{j \geq 1} \sum_{s > 1} \left(\frac{\delta}{40\pi^2 \cdot j^2}\right)^{2^s}$$

$$\leq \sum_{s > 1} \left(\frac{\delta}{8}\right)^{2^s}$$

$$\leq \frac{\delta}{16}. \tag{D.4}$$

Besides, from (D.3), we obtain

$$\mathbb{P}\left(\bigcap_{r \geq 1} \bigcap_{i \in [n]} \left\{|\hat{p}_i^r - \mu_i| \leq \frac{\epsilon_r}{8}\right\}\right) \geq 1 - \frac{\delta}{8}. \tag{D.5}$$

Define events

$$\mathcal{E}_3 = \cap_{r \geq 1} \cap_{i \in [n]} \left\{|\hat{p}_i^r - \mu_i| \leq \epsilon_r/8\right\},$$

and

$$\mathcal{E}_4 = \cap_{j \geq 1} \cap_{s > 1} \cap_{i \in S_j} \{|\hat{p}_i^{js} - \mu_i| < \epsilon_j/8\}.$$

If $\mathcal{E}_3$ and $\mathcal{E}_4$ truly hold, we have that (1): for any $j$ and arm $i \in S_j$,

$$\mu_i \leq \hat{p}_i^j + \frac{\epsilon_j}{8} \leq \hat{p}_*^j - \frac{7\epsilon_j}{8} \leq \mu_* - \frac{3\epsilon_j}{4} \leq \mu_1 - \frac{3\epsilon_j}{4},$$

where the second inequality is due to Line 11 of Algorithm 2; (2): for any fixed $j$, $s$, and any $r \in U_j(s)$, where $U_j(s)$ is defined in (C.2) and represents the set of all rounds in which parameter $\ell_j$ is updated for exactly $s$ times,

$$\hat{p}_i^{js} < \mu_i + \frac{\epsilon_j}{8} \leq \mu_1 - \frac{5\epsilon_j}{8} \leq \hat{p}_1^r - \frac{\epsilon_j}{2} \leq \hat{p}_*^r - \frac{\epsilon_j}{2}.$$

Therefore, if $\mathcal{E}_3$ and $\mathcal{E}_4$ truly hold, the event $\hat{p}_i^{js} < \hat{p}_*^r - \frac{\epsilon_j}{2}$ consistently holds for all $j$, $s$, $i \in S_j$, and $r \in U_j(s)$. It's noteworthy that, according to (D.4), $\mathbb{P}(\mathcal{E}_4) \geq 1 - \delta/16$, and from (D.5), $\mathbb{P}(\mathcal{E}_3) \geq 1 - \delta/8$. Therefore,

$$\mathbb{P}\left(\bigcap_{j \geq 1} \bigcap_{s > 1} \bigcap_{r \in U_j(s)} \left\{\max_{i \in S_j} \hat{p}_i^{js} < \hat{p}_*^r - \frac{\epsilon_j}{2}\right\}\right) \geq 1 - \frac{\delta}{8} - \frac{\delta}{16} = 1 - \frac{3\delta}{16}.$$

### D.3    Proof of Lemma C.3

We first focus on bounding the number of pulls of arm $i$ within the first **For Loop**. Define

$$r(i) = \min \left\{ r : \epsilon_r < \frac{\Delta_i}{2} \right\}.$$

From Lemma D.1 and the union bound, for $r \geq r(i)$,

$$\mathbb{P}\left( \left\{ |\hat{p}_i^r - \mu_i| \geq \frac{\epsilon_{r(i)}}{8} \right\} \cup \left\{ |\hat{p}_1^r - \mu_1| \geq \frac{\epsilon_{r(i)}}{8} \right\} \right) \leq 4 \exp\left( -\frac{2d_r \epsilon_{r(i)}^2}{64} \right)$$

$$\leq 4(\delta_r)^{r-r(i)}$$

$$\leq 10^{r(i)-r}. \tag{D.6}$$

Let $E_i^r$ be the event $|\hat{p}_i^r - \mu_i| \leq \epsilon_{r(i)}/8$ and $|\hat{p}_1^r - \mu_1| \leq \epsilon_{r(i)}/8$ truly hold at $r$-th round. Conditioned on $E_i^r$ and $\mathcal{E}$,

$$\hat{p}_i^r \leq \mu_i + \frac{\epsilon_r}{8} \leq \mu_1 - \Delta_i + \frac{\epsilon_r}{8} \leq \mu_1 - \frac{3\epsilon_r}{2} \leq \hat{p}_1^r - \epsilon_r \leq \hat{p}_*^r - \epsilon_r,$$

which mean arm $i$ will be eliminated. Therefore, the total sample cost of arm $i$ in the first **For Loop** of Algorithm 2 is

$$\sum_{r \leq r_i} \frac{32}{\epsilon_r^2} \log(2/\delta_r) + \sum_{r > r(i)} 10^{r-r(i)-1} \frac{32}{\epsilon_r^2} \log(2/\delta_r) = \mathcal{O}\left( \frac{\log(1/\delta_{r(i)})}{\epsilon_{r(i)}^2} \right)$$

$$= \mathcal{O}\left( \frac{\log\left( n/\delta \log(\Delta_i^{-1}) \right)}{\Delta_i^2} \right).$$

Consequently, if we define $H$ to be the total sample complexity in the first **For Loop** of Algorithm 2. Then we have

$$\mathbb{E}[H] = \mathcal{O}\left( \sum_{i>1} \frac{\log\left( n/\delta \log(\Delta_i^{-1}) \right)}{\Delta_i^2} \right).$$

Now, we bound the sample complexity within the second **For Loop**. We let $L_j$ be the total number of pulls of arms in $S_j \setminus S_{j+1}$ within the second **For Loop**. We let

$$n_j = \min \left\{ s \in \{0, 1, 2, \cdots\} : \frac{B_j}{(\delta_j)^{2^s} \cdot 2^s} \geq H \right\}.$$

As per Line 16, $n_j$ represents the total count of arm re-pulls in the set $S_j \setminus S_{j+1}$. Consequently, we can establish the following bound for $L_j$.

$$L_j \leq (B_j - B_{j-1}) \sum_{s=0}^{n_j} 2^s \leq B_j - B_{j-1} + \delta_j H.$$

where the first inequality is because $\gamma_{j(s+1)} = \gamma_{js}^2$ and we pull the arms in $S_j/S_{j+1}$ for total

$$\sum_{i \in S_j \setminus S_{j+1}} \frac{32}{\epsilon_j^2} \log\left( \frac{2}{\gamma_{js}} \right) \leq \sum_{i \in S_j \setminus S_{j+1}} \frac{32 \cdot 2^s}{\epsilon_j^2} \log\left( \frac{2}{\delta_j} \right) = (B_j - B_{j-1}) \cdot 2^s$$

times at Line 18 and the last inequality is because for $n_j > 1$,

$$\delta_j H \geq \delta_j \frac{B_j}{(\delta_j)^{2^{n_j-1}} \cdot 2^{n_j-1}} \geq B_j 2^{n_j+1},$$

where the first inequality is due to the definition of $n_j$. Therefore,

$$\sum_{j \geq 1} L_j \leq \sum_{j>1} B_j + H \sum_{j>1} \delta_j \leq 2H.$$

Now, we conclude that conditioned on event $\mathcal{E}$, the total expected sample complexity is

$$\mathcal{O}\left( \sum_{i>1} \frac{\log\left( n/\delta \log(\Delta_i^{-1}) \right)}{\Delta_i^2} \right).$$

## D.4 Proof of Lemma C.4

From Lemma D.1,

$$\mathbb{P}\left(|\hat{p}_i^r - \mu_i| \geq \frac{\epsilon_r}{2}\right) \leq 2\exp\left(-\frac{2d_r\epsilon_r^2}{4}\right) \leq (\delta_r)^4.$$

Applying union bound, we have

$$\mathbb{P}\left(\bigcup_{i\in S_r}\left\{|\hat{p}_i^r - \mu_i| \geq \frac{\epsilon_r}{2}\right\}\right) \leq \sum_{i\in S_r}(\delta_r)^4$$
$$\leq (\delta_r)^3. \tag{D.7}$$

The aforementioned inequality suggests that, with a probability of at least $1 - (\delta_r)^3$, the condition $|\hat{p}_i^r - \mu_i| \leq \epsilon_r/4$ consistently holds true for any $i \in S_r$. Then,

$$\hat{p}_1^r \geq \mu_1 - \frac{\epsilon_r}{2} \geq \mu_* - \frac{\epsilon_r}{2} \geq \hat{p}_*^r - \epsilon_r.$$

Therefore,

$$\mathbb{P}(\mathcal{E}^r, (\mathcal{E}^{r+1})^c) \leq (\delta_r)^3. \tag{D.8}$$

For any fixed $j$, recall the definition of $U_j(s)$ in (C.2). Assume $\mathcal{E}^r, (\mathcal{E}^{r+1})^c$ truly hold and denote $U_r(s) = \{r_s^1, r_s^2, \cdots, r_s^{s'}\}$ as the rounds where $\ell_r = s$. We note that at $r_s^l$-th round, each arm within the first **For Loop** of Algorithm 2 is pulled at least

$$\frac{32 \cdot 4^{s+l}}{\epsilon_r^2}\log\left(\frac{2}{\delta_r}\right)$$

times. Besides, each arm within the second **For Loop** is pulled at least

$$\frac{32 \cdot 2^s}{\epsilon_r^2}\log\left(\frac{2}{\delta_r}\right)$$

times. Then, similar to (D.7), for $r_s^l$-th round, we have

$$\mathbb{P}\left(\cup_{i\in S_{r_s^l}}\left\{|\hat{p}_i^{r_s^l} - \mu_i| \geq \frac{\epsilon_r}{4}\right\}\right) \leq 2\sum_{i\in S_{r_s^l}}\exp\left(\frac{-2\epsilon_r^2}{16} \cdot \frac{32 \cdot 4^{s+l}}{\epsilon_r^2}\log\left(\frac{2}{\delta_r}\right)\right) \leq \sum_{i\in S_{r_s^l}}(\delta_r^4)^{4^{s+l}}.$$

Then, we apply a union bound over all rounds in $U_r(s)$, we obtain

$$\mathbb{P}\left(\cup_{l\in[s']}\cup_{i\in S_{r_s^l}}\left\{|\hat{p}_i^{r_s^l} - \mu_i| \geq \frac{\epsilon}{4}\right\}\right) \leq (\delta_r^3)^{2^s}/2.$$

Moreover, if $\mathcal{E}^r, (\mathcal{E}^{r+1})^c$ happens, the best arm is eliminated at $r$-th round. Then for any rounds in $U_r(s)$, we have pulled arm 1 for at least

$$\frac{32 \cdot 2^s}{\epsilon_r^2}\log\left(\frac{2}{\delta_r}\right)$$

times in the second **For Loop**. Recall that $\hat{p}_i^{js}$ is the $s$th updates of parameter $\hat{p}_i^j$ within the second For Loop. From Lemma D.1, we have that

$$\mathbb{P}\left(\hat{p}_1^{rs} \leq \mu_1 - \frac{\epsilon_r}{4}\right) \leq \exp\left(-\frac{2\epsilon_r^2}{16} \cdot \frac{32 \cdot 2^s}{\epsilon_r^2}\log\left(\frac{2}{\delta_r}\right)\right) \leq (\delta_r^3)^{2^s}/2.$$

Therefore, if $\mathcal{E}^r, (\mathcal{E}^{r+1})^c$ happens, with probability

$$1 - \gamma_{rs} = 1 - (\delta_r^3)^{2^{s-1}}, \tag{D.9}$$

it holds that

$$\cup_{l\in[s']}\cup_{i\in S_{r_s^l}}\left\{|\hat{p}_i^{r_s^l} - \mu_i| \geq \frac{\epsilon}{4}\right\}, \text{ and } \hat{p}_1^{rs} > \mu_1 - \frac{\epsilon}{4},$$

and thus for all $l \in [s']$

$$\hat{p}_1^{rs} \geq \mu_1 - \frac{\epsilon}{4} \geq \hat{p}_*^{r_s^l} - \frac{\epsilon}{2},$$

which means Algorithm 2 returns at Line 22. Before we continue, we will first show that the number of pulls in the second **For Loop** is lower than the first **For Loop**. Assume the algorithm stops at $r'$-th round. Let $s_j' = \max\{s : B_{r'}\gamma_{js} \cdot 2^s > B_j\}$. The number of pulls for $S_j \setminus S_{j+1}$ at second **For Loop** is at most

$$
\begin{aligned}
\sum_{s=1}^{s_j'}(B_j - B_{j-1}) \cdot 2^s &\leq \sum_{s=1}^{s_j'} B_j 2^s \\
&\leq B_j \cdot 2^{s_j'+1} \\
&= 2^{2s_j'+1}\gamma_{js_j'}\frac{B_j}{2^{s_j'}\gamma_{js_j'}} \\
&\leq B_{r'}2^{2s_j'+1}(\delta_j)^{2^{s_j'}} \\
&\leq B_{r'}\delta_j.
\end{aligned}
$$

Therefore, the total number of pulls within the second **For Loop** is at most

$$\sum_{j\geq 1}\sum_{s=1}^{s_j'}(B_j - B_{j-1}) \cdot 2^s \leq B_{r'}\sum_{j>1}\delta_j \leq B_{r'},$$

which means the number of for the second **For Loop** is lower than the first **For Loop**. Finally, we obtain

$$
\begin{aligned}
\mathbb{E}[N'\mathbb{1}\{\mathcal{E}^r, (\mathcal{E}^{r+1})^c\}] &\leq \mathbb{E}\left(\mathcal{O}\left(\sum_{s=1}((\delta_r)^3)^{2^{s-1}} \cdot \frac{B_r}{(\delta_r)^{2^s}2^s}\right)\right) \\
&= \mathbb{E}[\mathcal{O}(\delta_r B_r)] \\
&= \mathcal{O}\left(\sum_{i>1}\frac{\delta_r \log\left(n/\delta\log(\Delta_i^{-1})\right)}{\Delta_i^2}\right).
\end{aligned}
$$

In the first inequality, we used the fact that if the algorithm stops in some rounds in $U_r(s)$, the total number of pulls of all arms is at most $B_r/(\gamma_{js} \cdot 2^s) + B_r/(\gamma_{js} \cdot 2^s)$, whcih comes from the first and second **For Loop** respectively. Moreover, the factor $((\delta_r)^3)^{2^{s-1}}$ is because from (D.9), if $\mathcal{E}^r, (\mathcal{E}^{r+1})^c$ holds, then Algorithm 2 returns in some rounds in $U_r(s)$ (at Line 22) with probability at least $1 - ((\delta_r)^3)^{2^{s-1}}$. The last equality is because from Lemma C.3, if $\mathcal{E}^r, (\mathcal{E}^{r+1})^c$ holds, then

$$\mathbb{E}[B_r] \leq \mathcal{O}\left(\sum_{i>1}\frac{\log\left(n/\delta\log(\Delta_i^{-1})\right)}{\Delta_i^2}\right).$$

Finally,

$$\sum_{r\geq 1}\mathbb{E}[N'\mathbb{1}\{\mathcal{E}^r, (\mathcal{E}^{r+1})^c\}] = \mathcal{O}\left(\sum_{i>1}\frac{\log\left(n/\delta\log(\Delta_i^{-1})\right)}{\Delta_i^2}\right).$$

### D.5 Proof of Lemma C.5

We note that each round within the **While Loop** costs one batch. Let

$$r(2) = \min\left\{r : \epsilon_r < \frac{\Delta_2}{2}\right\}.$$

Let $E^r$ be the event

$$\bigcap_{i>1}\left\{\left\{|\hat{p}_i^r - \mu_i| \leq \frac{\epsilon_{r(2)}}{8}\right\}\bigcap\left\{|\hat{p}_1^r - \mu_1| \leq \frac{\epsilon_{r(2)}}{8}\right\}\right\}.$$

Conditioned on $E^r$ and $\mathcal{E}$,
$$\hat{p}_i^r \le \mu_i + \frac{\epsilon_{r(2)}}{8} \le \mu_1 - \Delta_i - \frac{\epsilon_{r(2)}}{8} \le \hat{p}_1^r - \epsilon_{r(2)} \le \hat{p}_*^r - \epsilon_{r(2)},$$
which means all sub-optimal arms have been eliminated and the algorithm returns. From Lemma D.1 and the union bound, for $r \ge r(2)$,
$$\mathbb{P}((E^r)^c) \le 4n \exp\left(-\frac{2d_r \epsilon_{r(i)}^2}{64}\right) \le 4n(\delta_r)^{r-r(i)} \le 10^{r(i)-r}. \tag{D.10}$$
Therefore, the total batch cost is
$$r_2 + \sum_{r \ge r_2} \frac{1}{10^{r-r_2}} = O(r_2) = O\left(\log\left(\frac{1}{\Delta_2}\right)\right).$$
This completes the proof.

### D.6 Proof of Lemma C.6

The proof of this lemma is similar to that of Lemma C.4. we have the following results.
1: From (D.8), we have
$$\mathbb{P}(\mathcal{E}^r, (\mathcal{E}^{r+1})^c) \le (\delta_r)^3.$$
2: From (D.9), we have if $\mathcal{E}^r$ and $(\mathcal{E}^{r+1})^c$ truly hold, then with fixed $\ell_s$, Algorithm 2 returns at Line 22 with probability
$$1 - (\delta_r^3)^{2^{s-1}}. \tag{D.11}$$
We first compute the size of $U_r(s)$. From Algorithm 2, we know that the arm kept in the set $S_r$ will be pulled 4 times larger compared to $(r-1)$-th round. Besides, we update $\ell_r$ to $s+1$, if the number of pulls exceeds $B_r/((\delta_r)^{2^s} \cdot 2^s)$ (Line 16 of Algorithm 2). It is easy to see after $\log_4 n$ rounds, the number of pulls of any arm exceeds $B_r$ and then after $\ln \frac{1}{(\delta_r)^{2^s} \cdot 2^s}$ rounds, the number of pulls of single arm exceeds $B_r/((\delta_r)^{2^s} \cdot 2^s)$. Therefore, the size of $U_r(s)$ is at most $\ln \frac{1}{(\delta_r)^{2^s} \cdot 2^s} + \ln n$. We let $U$ be the total number of rounds used. Then, we obtain
$$\mathbb{E}[U \cdot \mathbb{1}\{\mathcal{E}^r, (\mathcal{E}^{r+1})^c\}] \le r \cdot \mathbb{P}(\mathcal{E}^r, (\mathcal{E}^{r+1})^c) + \mathbb{E}\left(\mathcal{O}\left(\sum_{s=1} ((\delta_r)^3)^{2^{s-1}} \cdot \left(\ln n + \ln \frac{1}{(\delta_r)^{2^s} \cdot 2^s}\right)\right)\right)$$
$$\le r\mathbb{P}(\mathcal{E}^r, (\mathcal{E}^{r+1})^c) + \delta_r.$$
We have shown in Lemma C.5, $\sum_{r>1} r\mathbb{P}(\mathcal{E}^r, (\mathcal{E}^{r+1})^c) = O(\log(1/\Delta_2))$. Therefore, the total number of rounds is
$$\sum_{r \ge 1}^{\infty} \mathbb{E}[U \cdot \mathbb{1}\{\mathcal{E}^r, (\mathcal{E}^{r+1})^c\}] = O(\log(1/\Delta_2)),$$
which completes the proof.

## E Experiments

In this section, we compare our algorithms Tri-BBAI and Opt-BBAI with Track-and-Stop [17], Top-k $\delta$-Elimination [22], ID-BAI [23] and CollabTopM [32] under bandit instances with Bernoulli rewards. All the experiments are repeated in 1000 trials. We perform all computations in Python on R9 5900HX for all our experiments. The implementation of this work can be found at https://github.com/panxulab/Optimal-Batched-Best-Arm-Identification

**Data generation.** For all experiments in this section, we set the number of arms $n = 10$, where each arm has Bernoulli reward distribution with mean $\mu_i$ for $i \in [10]$. More specifically, the mean rewards are generated by the following two cases.

- Uniform: The best arm has $\mu_1 = 0.5$, and the mean rewards of the rest of the arms follow uniform distribution over $[0.2, 0.4]$, i.e., $\mu_i$ is uniformly generated from $[0.2, 0.4]$ for $i \in [n] \setminus \{1\}$.
- Normal: The best arm has $\mu_1 = 0.6$, and the mean rewards of the rest of the arms are first generated from normal distribution $\mathcal{N}(0.2, 0.2)$ and then projected to the interval $[0, 0.4]$.

**Implementation details.**    The hyperparameters of all methods are chosen as follows.

- Track-and-Stop [17] is a fully sequential algorithm and thus the only parameter that needs to be set is the $\beta(t)$ function in the Chernoff's stopping condition (similar to Stage III of Algorithm 1). Note that the theoretical value of $\beta(t)$ in Track-and-Stop [17] is of the same order as presented in our Theorem 3.1. However, they found that a smaller value works better in practice. Therefore, we follow their experiments to set $\beta = \log\left((\log(t) + 1)/\delta\right)$.
- Top-k $\delta$-Elimination is a batched algorithm that eliminates the arms in batches. It has two parameters $\epsilon$ and $\delta$. In our experiments, we fix $\epsilon = 0.1$.
- ID-BAI [23] is designed to identify the best arm among a set of bandit arms with optimal instance-dependent sample complexity. We use the same algorithm setting of the original paper in our experiments.
- CollabTopM [32] is the algorithm to identify the Top-$m$ arms within a multi-agent setting. We set the $m$ as 1 and the agents $K$ as 1.
- For Tri-BBAI and Opt-BBAI, we set $\alpha = 1.001$[7], and $\epsilon = 0.01$. We use the same $\beta(t)$ function for Chernoff's stopping condition as in Track-and-Stop. Moreover, for the lengths of the batches, we set $L_1$, $L_2$ and $L3$ to be the value calculated by Theorem 3.1.

**Results.**    We present a comprehensive comparison on the sample complexities and batch complexities of our algorithms and baseline algorithms in Tables 3 and 4. Notably, our algorithms Tri-BBAI and Opt-BBAI, also including Top-K $\delta$ Elimination and CollabTopM, require significantly fewer batches than Track-and-Stop. Furthermore, the sample complexity of Tri-BBAI and Opt-BBAI is significantly lower than that of Top-K $\delta$ Elimination, ID-BAI and CollabTopM. Additionally, the sample complexity of Tri-BBAI and Opt-BBAI is comparable to Track and Stop when $\delta$ is large, and it is at most 3.6 times greater than Track and Stop when $\delta$ is very small. Additionally, we also provide the runtime comparison in Tables 3 and 4. Our algorithms have a significantly reduced runtime compared to Track-and-Stop, achieving nearly $1000\times$ speedup.

---

[7]From Theorem 3.1, we can select $\alpha$ to be any constant within the range $(1, e/2)$. To optimize the convergence rate of $\mathbb{E}[N_\delta]/(\log(1/\delta)T^*(\boldsymbol{\mu}))$, we set $\alpha$ slightly above 1, specifically we choose $\alpha = 1.001$.

Table 3: Experimental results in terms of sample complexity, batch complexity and runtime under the uniform mean rewards. The number of arms is $n = 10$. The experiments are averaged over 1000 repetitions.

| Dataset | $\delta$ | Algorithm | Sample Complexity | Batch Complexity | Runtime (s) | Recall |
|---|---|---|---|---|---|---|
| Uniform | $1 \times 10^{-1}$ | Track-and-Stop | $668.49 \pm 298.62$ | $668.49 \pm 298.62$ | $275.94 \pm 135.67$ | 100% |
| | | Top-k $\delta$-Elimination | $32472 \pm 0$ | $2 \pm 0.01$ | $0.04 \pm 0.01$ | 100% |
| | | ID-BAI | $120427 \pm 0$ | - | $0.136 \pm 0.003$ | 100% |
| | | CollabTopM | $27290.205 \pm 4806.435$ | $3.001 \pm 0.031$ | $0.032 \pm 0.006$ | 100% |
| | | Tri-BBAI | $365.232 \pm 21.85$ | $3.99 \pm 0.11$ | $1.02 \pm 0.23$ | 90.1% |
| | | Opt-BBAI | $1220.30 \pm 586.92$ | $4.89 \pm 1.01$ | $0.95 \pm 0.22$ | 100% |
| | $1 \times 10^{-2}$ | Track-and-Stop | $977.23 \pm 371.61$ | $977.23 \pm 371.61$ | $381.85 \pm 146.10$ | 100% |
| | | Top-k $\delta$-Elimination | $52734 \pm 0$ | $2 \pm 0.01$ | $0.07 \pm 0.01$ | 100% |
| | | ID-BAI | $146948 \pm 0$ | - | $0.167 \pm 0.003$ | 100% |
| | | CollabTopM | $35449.181 \pm 5530.938$ | $3.0 \pm 0$ | $0.042 \pm 0.006$ | 100% |
| | | Tri-BBAI | $1114.41 \pm 98.68$ | $3.96 \pm 0.19$ | $1.05 \pm 0.19$ | 99.3% |
| | | Opt-BBAI | $1929.65 \pm 604.11$ | $4.64 \pm 1.00$ | $1.02 \pm 0.18$ | 100% |
| | $1 \times 10^{-3}$ | Track-and-Stop | $1468.15 \pm 448.88$ | $1468.15 \pm 448.88$ | $624.82 \pm 219.14$ | 100% |
| | | Top-k $\delta$-Elimination | $72996 \pm 0$ | $2 \pm 0.01$ | $0.10 \pm 0.01$ | 100% |
| | | ID-BAI | $173478 \pm 0$ | - | $0.197 \pm 0.006$ | 100% |
| | | CollabTopM | $43457.489 \pm 5519.169$ | $3.0 \pm 0$ | $0.051 \pm 0.006$ | 100% |
| | | Tri-BBAI | $2005.07 \pm 175.13$ | $3.88 \pm 0.33$ | $1.09 \pm 0.13$ | 100% |
| | | Opt-BBAI | $2781.41 \pm 658.40$ | $4.37 \pm 0.98$ | $1.01 \pm 0.14$ | 100% |
| | $1 \times 10^{-4}$ | Track-and-Stop | $1769.72 \pm 467.36$ | $1769.72 \pm 467.36$ | $743.96 \pm 196.39$ | 100% |
| | | Top-k $\delta$-Elimination | $93258 \pm 0$ | $2 \pm 0.01$ | $0.12 \pm 0.01$ | 100% |
| | | ID-BAI | $199999 \pm 0$ | - | $0.257 \pm 0.005$ | 100% |
| | | CollabTopM | $51823.644 \pm 6379.803$ | $3.0 \pm 0$ | $0.061 \pm 0.007$ | 100% |
| | | Tri-BBAI | $2989.96 \pm 274.36$ | $3.81 \pm 0.39$ | $1.10 \pm 0.13$ | 100% |
| | | Opt-BBAI | $3752.31 \pm 739.60$ | $4.20 \pm 0.99$ | $1.00 \pm 0.12$ | 100% |
| | $1 \times 10^{-5}$ | Track-and-Stop | $2135.09 \pm 535.01$ | $2135.09 \pm 535.01$ | $821.59 \pm 137.83$ | 100% |
| | | Top-k $\delta$-Elimination | $113520 \pm 0$ | $2 \pm 0.01$ | $0.15 \pm 0.01$ | 100% |
| | | ID-BAI | $226528 \pm 0$ | - | $0.257 \pm 0.005$ | 100% |
| | | CollabTopM | $60263.887 \pm 6367.285$ | $3.0 \pm 0$ | $0.071 \pm 0.007$ | 100% |
| | | Tri-BBAI | $4066.34 \pm 381.41$ | $3.74 \pm 0.44$ | $1.11 \pm 0.13$ | 100% |
| | | Opt-BBAI | $4799.49 \pm 852.00$ | $4.04 \pm 0.96$ | $1.03 \pm 0.12$ | 100% |
| | $1 \times 10^{-6}$ | Track-and-Stop | $2517.93 \pm 561.65$ | $2517.93 \pm 561.65$ | $1085.30 \pm 202.72$ | 100% |
| | | Top-k $\delta$-Elimination | $133782 \pm 0$ | $2 \pm 0.01$ | $0.18 \pm 0.01$ | 100% |
| | | ID-BAI | $253049 \pm 0$ | - | $0.288 \pm 0.006$ | 100% |
| | | CollabTopM | $67958.078 \pm 6730.415$ | $3.0 \pm 0$ | $0.081 \pm 0.008$ | 100% |
| | | Tri-BBAI | $5185.75 \pm 485.82$ | $3.65 \pm 0.48$ | $1.12 \pm 0.12$ | 100% |
| | | Opt-BBAI | $5871.33 \pm 951.60$ | $3.90 \pm 0.93$ | $1.07 \pm 0.14$ | 100% |
| | $1 \times 10^{-7}$ | Track-and-Stop | $2942.67 \pm 598.77$ | $2942.67 \pm 598.77$ | $1232.91 \pm 192.82$ | 100% |
| | | Top-k $\delta$-Elimination | $154044 \pm 0$ | $2 \pm 0.01$ | $0.21 \pm 0.01$ | 100% |
| | | ID-BAI | $279579 \pm 0$ | - | $0.317 \pm 0.006$ | 100% |
| | | CollabTopM | $76627.155 \pm 7000.211$ | $3.0 \pm 0$ | $0.090 \pm 0.008$ | 100% |
| | | Tri-BBAI | $6334.49 \pm 613.88$ | $3.58 \pm 0.49$ | $1.12 \pm 0.12$ | 100% |
| | | Opt-BBAI | $7055.16 \pm 1059.50$ | $3.82 \pm 0.93$ | $1.11 \pm 0.16$ | 100% |
| | $1 \times 10^{-8}$ | Track-and-Stop | $3347.02 \pm 535.16$ | $3347.02 \pm 535.16$ | $1464.84 \pm 359.57$ | 100% |
| | | Top-k $\delta$-Elimination | $174306 \pm 0$ | $2 \pm 0.01$ | $0.23 \pm 0.01$ | 100% |
| | | ID-BAI | $306108 \pm 0$ | - | $0.347 \pm 0.006$ | 100% |
| | | CollabTopM | $84995.542 \pm 7270.987$ | $3.0 \pm 0$ | $0.100 \pm 0.009$ | 100% |
| | | Tri-BBAI | $7486.57 \pm 764.36$ | $3.49 \pm 0.50$ | $1.13 \pm 0.11$ | 100% |
| | | Opt-BBAI | $8136.01 \pm 1094.05$ | $3.64 \pm 0.85$ | $1.10 \pm 0.12$ | 100% |
| | $1 \times 10^{-9}$ | Track-and-Stop | $3609.64 \pm 638.68$ | $3609.64 \pm 638.68$ | $1661.96 \pm 266.15$ | 100% |
| | | Top-k $\delta$-Elimination | $194568 \pm 0$ | $2 \pm 0.01$ | $0.26 \pm 0.01$ | 100% |
| | | ID-BAI | $332628 \pm 0$ | - | $0.376 \pm 0.006$ | 100% |
| | | CollabTopM | $92705.663 \pm 7639.167$ | $3.0 \pm 0$ | $0.109 \pm 0.009$ | 100% |
| | | Tri-BBAI | $8759.03 \pm 844.59$ | $3.41 \pm 0.49$ | $1.12 \pm 0.10$ | 100% |
| | | Opt-BBAI | $9315.58 \pm 1264.28$ | $3.56 \pm 0.83$ | $1.01 \pm 0.10$ | 100% |
| | $1 \times 10^{-10}$ | Track-and-Stop | $4136.94 \pm 665.10$ | $4136.94 \pm 665.10$ | $1714.68 \pm 257.00$ | 100% |
| | | Top-k $\delta$-Elimination | $214830 \pm 0$ | $2 \pm 0.01$ | $0.28 \pm 0.01$ | 100% |
| | | ID-BAI | $359158 \pm 0$ | - | $0.407 \pm 0.008$ | 100% |
| | | CollabTopM | $100894.264 \pm 7704.855$ | $3.0 \pm 0$ | $0.119 \pm 0.009$ | 100% |
| | | Tri-BBAI | $10083.60 \pm 1005.54$ | $3.30 \pm 0.45$ | $1.26 \pm 0.18$ | 100% |
| | | Opt-BBAI | $10548.19 \pm 1277.15$ | $3.48 \pm 0.78$ | $0.99 \pm 0.09$ | 100% |

Table 4: Experimental results in terms of sample complexity, batch complexity and runtime under the normal mean rewards. The number of arms is $n = 10$. The experiments are averaged over 1000 repetitions.

| Dataset | $\delta$ | Algorithm | Sample Complexity | Batch Complexity | Runtime (s) | Recall |
|---|---|---|---|---|---|---|
| Normal | $1 \times 10^{-1}$ | Track-and-Stop | $305.21 \pm 183.51$ | $305.21 \pm 183.51$ | $154.42 \pm 64.49$ | 100% |
| | | Top-k $\delta$-Elimination | $32472.0 \pm 0$ | $2 \pm 0.01$ | $0.04 \pm 0.01$ | 100% |
| | | ID-BAI | $120427 \pm 0$ | - | $0.145 \pm 0.012$ | 100% |
| | | CollabTopM | $11834.089 \pm 2208.691$ | $2.888 \pm 0.315$ | $0.015 \pm 0.003$ | 100% |
| | | Tri-BBAI | $334.49 \pm 32.80$ | $3.89 \pm 0.32$ | $1.08 \pm 0.16$ | 98.3% |
| | | Opt-BBAI | $793.23 \pm 358.15$ | $4.18 \pm 0.79$ | $1.05 \pm 0.16$ | 100% |
| | $1 \times 10^{-2}$ | Track-and-Stop | $490.95 \pm 206.28$ | $490.95 \pm 206.28$ | $182.42 \pm 81.61$ | 100% |
| | | Top-k $\delta$-Elimination | $52734 \pm 0$ | $2 \pm 0.01$ | $0.07 \pm 0.01$ | 100% |
| | | ID-BAI | $146948 \pm 0$ | - | $0.171 \pm 0.011$ | 100% |
| | | CollabTopM | $16145.218 \pm 2255.120$ | $2.933 \pm 0.250$ | $0.021 \pm 0.004$ | 100% |
| | | Tri-BBAI | $893.05 \pm 121.47$ | $3.62 \pm 0.48$ | $1.11 \pm 0.11$ | 100% |
| | | Opt-BBAI | $1236.33 \pm 381.17$ | $3.74 \pm 0.71$ | $1.11 \pm 0.13$ | 100% |
| | $1 \times 10^{-3}$ | Track-and-Stop | $699.18 \pm 198.64$ | $699.18 \pm 198.64$ | $264.66 \pm 116.82$ | 100% |
| | | Top-k $\delta$-Elimination | $72996 \pm 0$ | $2 \pm 0.01$ | $0.10 \pm 0.01$ | 100% |
| | | ID-BAI | $173478 \pm 0$ | - | $0.204 \pm 0.013$ | 100% |
| | | CollabTopM | $20414.549 \pm 2277.714$ | $2.960 \pm 0.195$ | $0.027 \pm 0.004$ | 100% |
| | | Tri-BBAI | $1532.76 \pm 201.18$ | $3.37 \pm 0.48$ | $1.13 \pm 0.10$ | 100% |
| | | Opt-BBAI | $1747.15 \pm 389.65$ | $3.43 \pm 0.58$ | $1.07 \pm 0.10$ | 100% |
| | $1 \times 10^{-4}$ | Track-and-Stop | $833.08 \pm 234.48$ | $833.08 \pm 231.90$ | $315.10 \pm 91.15$ | 100% |
| | | Top-k $\delta$-Elimination | $93258 \pm 0$ | $2 \pm 0.01$ | $0.12 \pm 0.01$ | 100% |
| | | ID-BAI | $199999 \pm 0$ | - | $0.236 \pm 0.015$ | 100% |
| | | CollabTopM | $24673.111 \pm 2584.898$ | $2.962 \pm 0.191$ | $0.032 \pm 0.005$ | 100% |
| | | Tri-BBAI | $2141.11 \pm 282.37$ | $3.20 \pm 0.40$ | $1.13 \pm 0.08$ | 100% |
| | | Opt-BBAI | $2263.174 \pm 405.72$ | $3.19 \pm 0.42$ | $1.06 \pm 0.08$ | 100% |
| | $1 \times 10^{-5}$ | Track-and-Stop | $972.75 \pm 245.18$ | $972.75 \pm 245.18$ | $336.78 \pm 74.03$ | 100% |
| | | Top-k $\delta$-Elimination | $113520 \pm 0$ | $2 \pm 0.01$ | $0.15 \pm 0.01$ | 100% |
| | | ID-BAI | $226528 \pm 0$ | - | $0.269 \pm 0.044$ | 100% |
| | | CollabTopM | $29081.141 \pm 2323.105$ | $2.982 \pm 0.132$ | $0.039 \pm 0.005$ | 100% |
| | | Tri-BBAI | $2838.36 \pm 353.12$ | $3.09 \pm 0.28$ | $1.14 \pm 0.08$ | 100% |
| | | Opt-BBAI | $2881.61 \pm 430.05$ | $3.08 \pm 0.27$ | $1.09 \pm 0.09$ | 100% |
| | $1 \times 10^{-6}$ | Track-and-Stop | $1122.53 \pm 308.73$ | $1122.53 \pm 308.73$ | $468.14 \pm 138.87$ | 100% |
| | | Top-k $\delta$-Elimination | $133782 \pm 0$ | $2 \pm 0.01$ | $0.17 \pm 0.01$ | 100% |
| | | ID-BAI | $253049 \pm 0$ | - | $0.312 \pm 0.038$ | 100% |
| | | CollabTopM | $33386.52 \pm 2133.062$ | $2.987 \pm 0.113$ | $0.043 \pm 0.005$ | 100% |
| | | Tri-BBAI | $3516.52 \pm 467.48$ | $3.03 \pm 0.18$ | $1.14 \pm 0.08$ | 100% |
| | | Opt-BBAI | $3556.59 \pm 477.73$ | $3.04 \pm 0.20$ | $1.15 \pm 0.15$ | 100% |
| | $1 \times 10^{-7}$ | Track-and-Stop | $1256.29 \pm 308.88$ | $1256.29 \pm 308.88$ | $544.76 \pm 74.00$ | 100% |
| | | Top-k $\delta$-Elimination | $154044 \pm 0$ | $2 \pm 0.01$ | $0.20 \pm 0.02$ | 100% |
| | | ID-BAI | $279579 \pm 0$ | - | $0.336 \pm 0.019$ | 100% |
| | | CollabTopM | $37499.802 \pm 2413.418$ | $2.986 \pm 0.117$ | $0.048 \pm 0.005$ | 100% |
| | | Tri-BBAI | $4218.39 \pm 523.49$ | $3.01 \pm 0.09$ | $1.15 \pm 0.08$ | 100% |
| | | Opt-BBAI | $4220.78 \pm 523.11$ | $3.02 \pm 0.13$ | $1.16 \pm 0.13$ | 100% |
| | $1 \times 10^{-8}$ | Track-and-Stop | $1438.20 \pm 355.44$ | $1438.20 \pm 355.44$ | $566.44 \pm 101.01$ | 100% |
| | | Top-k $\delta$-Elimination | $174306 \pm 0$ | $2 \pm 0.01$ | $0.23 \pm 0.01$ | 100% |
| | | ID-BAI | $306108 \pm 0$ | - | $0.362 \pm 0.018$ | 100% |
| | | CollabTopM | $41924.698 \pm 2206.559$ | $2.992 \pm 0.089$ | $0.055 \pm 0.005$ | 100% |
| | | Tri-BBAI | $4912.64 \pm 613.69$ | $3.01 \pm 0.08$ | $1.15 \pm 0.07$ | 100% |
| | | Opt-BBAI | $4940.68 \pm 650.75$ | $3.00 \pm 0.06$ | $1.12 \pm 0.10$ | 100% |
| | $1 \times 10^{-9}$ | Track-and-Stop | $1632.12 \pm 348.14$ | $1632.12 \pm 348.14$ | $590.42 \pm 56.25$ | 100% |
| | | Top-k $\delta$-Elimination | $194568 \pm 0$ | $2 \pm 0.01$ | $0.26 \pm 0.01$ | 100% |
| | | ID-BAI | $332628 \pm 0$ | - | $0.393 \pm 0.017$ | 100% |
| | | CollabTopM | $46244.241 \pm 1816.041$ | $2.997 \pm 0.054$ | $0.059 \pm 0.005$ | 100% |
| | | Tri-BBAI | $5637.36 \pm 743.53$ | $3.00 \pm 0.05$ | $1.14 \pm 0.08$ | 100% |
| | | Opt-BBAI | $5661.51 \pm 740.74$ | $3.00 \pm 0.05$ | $1.03 \pm 0.06$ | 100% |
| | $1 \times 10^{-10}$ | Track-and-Stop | $1747.29 \pm 318.85$ | $1747.29 \pm 318.85$ | $790.89 \pm 119.52$ | 100% |
| | | Top-k $\delta$-Elimination | $214830 \pm 0$ | $2 \pm 0.01$ | $0.28 \pm 0.01$ | 100% |
| | | ID-BAI | $359158 \pm 0$ | - | $0.423 \pm 0.015$ | 100% |
| | | CollabTopM | $50556.41 \pm 1822.998$ | $2.997 \pm 0.054$ | $0.067 \pm 0.007$ | 100% |
| | | Tri-BBAI | $6356.78 \pm 787.53$ | $3.00 \pm 0.03$ | $1.30 \pm 0.16$ | 100% |
| | | Opt-BBAI | $6355.28 \pm 793.45$ | $3.00 \pm 0.03$ | $1.03 \pm 0.05$ | 100% |

