# OpenReview forum: "Optimal Batched Best Arm Identification"
_NeurIPS.cc/2024/Conference — NeurIPS 2024 poster_

### Official Review · Reviewer_Uhz8 · 2024-07-01

**Soundness:** 3
**Presentation:** 3
**Contribution:** 2
**Rating:** 6
**Confidence:** 4

**Summary:**

This paper studies the Batched Best Arm Identification (BBAI) problem in multi-armed bandits. The goal is to design an efficient algorithm that correctly finds the arm with the highest mean with probability $\ge 1 - \delta$, while minimizing: (1) the sample complexity, defined as the total number of arm pulls; (2) the "batch complexity", defined as the number of rounds in which the algorithm requests arm pulls. (Formally, the algorithm specifies the number of times it pulls each arm in the $r$-th round after observing all the outcomes in the previous $r-1$ rounds.)

The main results of this work are:
- An algorithm (termed Tri-BBAI) that achieves an asymptotically (i.e., as $\delta \to 0^{+}$) optimal sample complexity in three batches in expectation.
- Another algorithm (termed Opt-BBAI) that achieves: (1) near-optimal sample and batch complexities in the non-asymptotic setting; (2) the same guarantees as Tri-BBAI in the asymptotic setting.

**Strengths:**

- This work studies a fairly natural problem. While the setup is not new, I liked that the authors explored certain perspectives that are different from prior work, namely: (1) asymptotic optimality of sample complexity; (2) focusing on the expected number of rounds, rather than treating it as a hard constraint.
- The results are pretty strong: The algorithms achieve asymptotically optimal sample complexities within a (small) constant number of rounds (in expectation).
- The presentation is clear in general. I found the main paper fairly easy to follow.

**Weaknesses:**

I think a major weakness of the work is that its main result is fairly intuitive, and arguably, its proof does not give new insights to the design of pure exploration algorithms. Here is why: For simplicity, suppose that we only have two arms with means $1/2+\epsilon$ and $1/2-\epsilon$, and the parameter $\epsilon > 0$ is *unknown*.

In this case, the "natural" algorithm is to make geometrically decreasing guesses on $\epsilon$: $\epsilon_1, \epsilon_2, \ldots$, where each $\epsilon_k = 2^{-k}$. At guess $\epsilon_k$, we would pull each arm $(1/\epsilon_k)^2$ times to get an $O(\epsilon_k)$-approximation of the means. If the means are clearly separated, we get the answer; otherwise, we continue with smaller guesses. Clearly, this approach requires many rounds ($\Omega(\log(1/\epsilon))$ rounds) of adaptivity.

On this instance, very roughly speaking, the authors' approach is to start with the guess, say, $\epsilon_k = 1/\log^{1/3}(1/\delta)$. For each fixed $\epsilon$, when $\delta$ is small enough, the guess would be smaller than the actual $\epsilon$, and the algorithm wins the game in $O(1)$ rounds. Also, this first round only takes $(1/\epsilon_k)^2 = \log^{2/3}(1/\delta) \ll \log(1/\delta)$ samples, so this will not affect the asymptotic behavior as $\delta \to 0^{+}$.

In general, the Tri-BBAI algorithm uses a round-robin strategy as an inefficient exploration round. The length of this round is chosen as a function of $\delta$, so that the asymptotic behavior is not affected. Therefore, for each fixed instance, there exists some $\delta_0 > 0$ such that whenever $\delta < \delta_0$, this inefficient exploration succeeds with a good probability, and may guide the algorithm to sample in an asymptotically optimal way in the rest. As a result, this asymptotic optimality might only hold for extremely small values of $\delta$: From Line 492, the analysis needs $1/\log\log(1/\delta) = \epsilon \le \Delta_2$ to go through; in other words, $1/\delta$ needs to be **doubly exponential** in $1/\Delta_2$.

Admittedly, this weakness was addressed by the other algorithm Opt-BBAI, which achieves a sample complexity bound for finite $\delta$ as well. However, it should be noted that the complexity contains a $\sum_{i=2}^{n}(1/\Delta_i)^2\log n$ term, which could be higher than the optimal sample complexity by a $\log n$ factor. In contrast, the state-of-the-art bounds for (non-batched) BAI (e.g., [Karnin-Koren-Somekh, ICML'13][Jamieson-Malloy-Nowak-Bubeck, COLT'14][Chen-Li, arXiv'15][Chen-Li-Qiao, COLT'17]) are tight up to a doubly-logarithmic factor.

Despite the weakness mentioned above, I think this submission presents some solid work and a nice observation (Lines 72-73), namely, the need for many rounds of adaptivity only arises when $\delta$ is small (or moderate), and goes away when we focus on the asymptotic regime. Therefore, I lean towards accepting the paper.

**Questions:**

I don't have specific questions for the authors.

**Limitations:**

Limitations have been adequately addressed.

---

> ### Author Rebuttal · Authors · 2024-08-07
>
> Thank you for your encouraging comments. The example you provided perfectly captures our main idea. We greatly appreciate this clear and concise explanation.

---

### Official Review · Reviewer_vkzZ · 2024-07-07

**Soundness:** 2
**Presentation:** 2
**Contribution:** 1
**Rating:** 3
**Confidence:** 4

**Summary:**

The paper "Optimal Batched Best Arm Identification" introduces Tri-BBAI and Opt-BBAI algorithms to identify the best arm in multi-armed bandit settings. Tri-BBAI achieves optimal sample complexity with only three batches on average as the confidence parameter $\delta$ approaches zero. Opt-BBAI extends Tri-BBAI for finite-confidence settings, providing near-optimal sample and batch complexities for finite $\delta$.

**Strengths:**

The paper is generally well-written. The theoretical analysis looks solid.

**Weaknesses:**

First, I don't think the regime when $\delta$ approaches $0$ is interesting. In a typical scenario, we want to set the confidence relatively small, and it doesn't make a lot of sense to consider the algorithm's performance on extremely small values.

The paper doesn't fairly compare previous results. It mentions the work "Collaborative top distribution identifications with limited interaction," and work "Optimal streaming algorithms for multi-armed bandits", but it doesn't include it in the table, which can lead to the impression that Opt-BBAI is the first algorithm to achieve $O(\log \frac{1}{\Delta_{2}})$ batch complexity. However, many algorithms have already achieved similar results. Moreover, these algorithms are not included in experiments. Significant work should be done to implement a fair comparison and properly acknowledge other works.

**Questions:**

N/A

---

> ### Author Rebuttal · Authors · 2024-08-07
>
> We thank the reviewer for your valuable time and effort in providing detailed feedback on our work. We hope our response will fully address all your questions.
>
> ---
> Q1: I don't think the regime when $\delta$ approaches 0 is interesting. In a typical scenario, we want to set the confidence relatively small, and it doesn't make a lot of sense to consider the algorithm's performance on extremely small values.
>
> A1: We respectfully disagree with this argument for the following reasons:
>
> 1. Many influential works adopt asymptotic optimality and have been published in top conferences such as ICML, NeurIPS, AISTATS, and COLT. We only mention a few here, including:
>
> [1] Garivier, Aurélien, and Emilie Kaufmann. "Optimal best arm identification with fixed confidence." COLT 2016.
>
> [2] Degenne, Rémy, et al. "Gamification of pure exploration for linear bandits." ICML 2020.
>
> [3] Jedra, Yassir, and Alexandre Proutiere. "Optimal best-arm identification in linear bandits." NeurIPS 2020.
>
> [4] Lattimore, Tor, and Csaba Szepesvari. "The end of optimism? an asymptotic analysis of finite-armed linear bandits." AISTATS 2017.
>
> [5] Degenne, Rémy, Wouter M. Koolen, and Pierre Ménard. "Non-asymptotic pure exploration by solving games." NeurIPS 2019.
>
> [6] Kirschner, Johannes, et al. "Asymptotically optimal information-directed sampling." COLT 2021.
>
> In recent years:
>
> [7] Jourdan, Marc, and Rémy Degenne. "Non-asymptotic analysis of a UCB-based top two algorithm." NeurIPS 2023.
>
> [8] You, Wei, et al. "Information-directed selection for top-two algorithms." COLT 2023.
>
> [9] Jourdan, Marc, Rémy Degenne, and Emilie Kaufmann. "An $\epsilon$-Best-Arm Identification Algorithm for Fixed-Confidence and Beyond." NeurIPS 2023.
>
> [10] Jourdan, Marc, et al. "Top two algorithms revisited." NeurIPS 2022.
>
> [11] Deep, Vikas, Achal Bassamboo, and Sandeep Kumar Juneja. "Asymptotically Optimal and Computationally Efficient Average Treatment Effect Estimation in A/B testing." ICML 2024.
>
> [12] Wang, Po-An, Kaito Ariu, and Alexandre Proutiere. "On Universally Optimal Algorithms for A/B Testing." ICML 2024.
>
> [13] Ren, Xuanfei, Tianyuan Jin, and Pan Xu. "Optimal Batched Linear Bandits." ICML 2024.
>
> 2. Considering the asymptotic performance is also practical. An algorithm that is optimal as $\delta$ approaches 0 consistently shows good empirical performance compared to algorithms that involve large constants in sample complexity. The experimental results in this paper, along with comparisons to the two extra works you suggested (see Table 1 in the general response and Table 2 in the attached PDF file), all demonstrate this fact.
>
> 3. Our paper also contributes to the field of batched algorithms in the non-asymptotic setting. As highlighted in our contributions (Lines 94-100), the batch/sample complexity of earlier batch algorithms typically depends on the event of returning the best arm, which occurs with a probability of at least $1−\delta$. However, this complexity could potentially become unbounded if a sub-optimal arm is returned instead. In contrast, the complexity of Opt-BBAI is not contingent on such an event. This is achieved through a new technique, i.e., Checking for Best Arm Elimination (see Lines 277-288).
>
> ---
> Q2: The paper mentions the work [14] and [15], doesn't include it in the table, which can lead to the impression that Opt-BBAI is the first algorithm to achieve $O(\log⁡ (1/\Delta_2))$ batch complexity.
>
> A2: For [15],  their algorithm is actually not for the batched setting. All their algorithms call their Algorithm 1 as a component which is fully sequential. Consequently, their algorithm is not for the batch setting.
>
> For [14], we reported their results in Lines 151 and 152. In the revision, we will include it in the table as well. Regarding your comment that “which can lead to the impression that Opt-BBAI is the first algorithm to achieve $O(\log(1/\Delta_2))$ batch complexity.”, we believe this is a misunderstanding of our work since
> 1. We explicitly mentioned in Lines 151 and 152 that the algorithm in [14] runs in $O(\log (1/\Delta_2))$ batches. We will add their batch complexity to our table in the revision.
> 2. We did not claim to have developed the first algorithm to achieve $O(\log(1/\Delta_2))$ batch complexity. Instead, as clearly outlined in Lines 86-101, our primary contribution is that we propose the first algorithm that adaptively achieves optimal sample and batch complexity in an asymptotic setting, and near-optimal sample and batch complexity in a non-asymptotic setting. Additionally, we have introduced new techniques for addressing issues in previous work, where batch and sample complexity depended on high-probability events.
>
> We will polish our writing further to incorporate the above discussions into our revision for a clearer presentation.
>
> ---
> Q3: Experimental comparison with [14] and [15].
>
> A3: We added these two baselines. Due to space limitations, the experimental results are shown in the general response and the attached PDF file. The experimental results demonstrate that our sample complexity is significantly lower than that of the two baselines. This is because our algorithms are asymptotically optimal, whereas the sample complexity in the baselines involves large constants. We will incorporate these results in the final version.
>
> ---
> We hope we have addressed all of your questions/concerns. If you have any further questions, we would be more than happy to answer them and if you don’t, would you kindly consider increasing your score?
>
> ---
> **References**
>
> [14] Karpov, N., Zhang, Q. and Zhou, Y., 2020, November. Collaborative top distribution identifications with limited interaction. In 2020 IEEE 61st Annual Symposium on Foundations of Computer Science (FOCS) (pp. 160-171). IEEE.
>
> [15] Jin, T., Huang, K., Tang, J. and Xiao, X., 2021, July. Optimal streaming algorithms for multi-armed bandits. In International Conference on Machine Learning (pp. 5045-5054). PMLR.

---

> > ### Author Response · Authors · 2024-08-12
> >
> > Dear Reviewer vkzZ,
> >
> > Thanks again for reviewing our paper. As the end of the discussion period approaches, we would like to know whether our responses have addressed your concerns. If there are any additional questions or areas that require clarification, please do not hesitate to let us know. We highly value your perspective and, if you find our responses satisfactory, would be grateful if you would consider raising your score for our paper.
> >
> > Thank you.

---

> > > ### Comment · Reviewer_vkzZ · 2024-08-14
> > >
> > > Thank you for your feedback; I will keep my score as it is.

---

> > > > ### Author Response · Authors · 2024-08-14
> > > >
> > > > Thank you for your reply. We would like to summarize our responses to your review as follows:
> > > >
> > > > 1. You raised concerns that the asymptotic setting (as $\delta$ approaches 0) is not interesting.
> > > > 2. You noted the omission of two related works (‘Collaborative Top Distribution Identification with Limited Interaction’ and ‘Optimal Streaming Algorithms for Multi-Armed Bandits’) from our comparison table.
> > > >
> > > > In our previous response, we addressed these points, which we believe are relatively minor and do not detract from the overall significance of our technical contributions. Specifically:
> > > >
> > > > * We demonstrated that the asymptotic setting is not only widely studied in the literature but also holds practical significance (see A1 for more details).
> > > > * We emphasized our substantial contributions to the non-asymptotic setting (finite $\delta$), as discussed in both our paper and the previous response.
> > > > * We clarified that 'Optimal Streaming Algorithms for Multi-Armed Bandits' is not designed for  the batched bandit setting, and that we have already cited the main results of 'Collaborative Top Distribution Identification with Limited Interaction' in Lines 151 and 152 of our paper.
> > > > * We committed to add more discussions on these papers and add related results in our comparison table and we have conducted additional experiments in the rebuttal which will be included in the revised manuscript.
> > > >
> > > > We hope this summary clarifies our position and the steps we have taken to address your concerns.
> > > >
> > > > Best regards,
> > > >
> > > > Authors

---

### Official Review · Reviewer_3W3Z · 2024-07-13

**Soundness:** 3
**Presentation:** 3
**Contribution:** 3
**Rating:** 7
**Confidence:** 3

**Summary:**

This paper considers the problem of BAI in the fixed confidence setting, in the context of finding the best arm in as few batches as possible. That is, instead of observing the reward of each arm pulled in turn, the learner makes multiple pulls in a single batch and only observes all rewards after the completion of said batch. The first result of the paper is to show that it is possible to achieve asymptotically optimal sample complexity, with only a constant number of batches. Specifically the authors propose the Tri-BBAI algorithm, which is shown to have asymptotically optimal sample complexity, PAC guarantees and requiring at most 3 batches in expectation, Theorems 3.1, 3.2, 3.3 respectively.

The authors then describe a second algorithm Opt-BBAI which has near optimal finite confidence guarantees as well as being asymptotically optimal.

**Strengths:**

Best arm identification under fixed confidence is well studied problem and running algorithms in batches has clear practical relevance. Thus, achieving both asymptotically optimal sample and constant batch complexity is a nice result.

**Weaknesses:**

When considering finite confidence guarantees the authors could also compare with the recent work "An ε-Best-Arm Identification Algorithm for Fixed-Confidence and Beyond", Jourdan, Degenne, Kaufmann.

**Questions:**

Do the authors have an idea as to why the Tri-BBAI has poor experimental performace against Track and Stop?

**Limitations:**

No concern.

---

> ### Author Rebuttal · Authors · 2024-08-07
>
> We thank the reviewer for their valuable time and effort in providing detailed feedback on our work. We hope our response will fully address all of the reviewer's points.
>
> ---
>
> Q1: When considering finite confidence guarantees the authors could also compare with the recent work "An $\epsilon$-Best-Arm Identification Algorithm for Fixed-Confidence and Beyond", Jourdan, Degenne, Kaufmann.
>
> A1: Thank you for highlighting this reference. The paper studies $\epsilon$-best arm identification, proposing an asymptotically optimal algorithm and providing non-asymptotic sample complexity. [1] is not for the batched setting. When $\epsilon = 0$, it aligns with our setting, and our finite sample complexity is better scaled. Specifically, [1] offers a non-asymptotic sample complexity scale of $n/\Delta_{2}^2 \log (1/\Delta_2)$, whereas ours is more instance-sensitive, as our sample complexity is related to all gaps, not just $\Delta_2$. Additionally, [1] considers a practical scenario where the algorithm can return a result at any time while still ensuring a good guarantee on the returned arm. We will compare our results with those in [1] in our revision.
>
> ---
> **Reference**
>
> [1] Jourdan M, Degenne R, Kaufmann E. An $\varepsilon $-Best-Arm Identification Algorithm for Fixed-Confidence and Beyond[J]. Advances in Neural Information Processing Systems, 2023, 36: 16578-16649.

---

> > ### Author Response · Authors · 2024-08-12
> >
> > Dear Reviewer 3W3Z,
> >
> > We would like to know whether our responses have addressed your concerns. If there are any additional questions or areas that require clarification, please do not hesitate to let us know.
> >
> > Thank you.

---

> > > ### Comment · Reviewer_3W3Z · 2024-08-13
> > >
> > > Thank you for your response, I will maintain my score.

---

### Official Review · Reviewer_YCPj · 2024-07-13

**Soundness:** 3
**Presentation:** 3
**Contribution:** 3
**Rating:** 5
**Confidence:** 2

**Summary:**

The paper presents two novel algorithms for the batched best arm identification (BBAI) problem. The first is the Tri-BBAI algorithm, which employs three batches with the expectation of achieving the asymptotic optimal sample complexity. Based on Tri-BBAI, the authors conceived the Opt-BBAI algorithm, which achieves the near-optimal sample and batch complexity when δ is finite and it does not tend to 0. Opt-BBAI enjoys the same sample and batch complexity of Tri-BBAI for the asymptotic setting

**Strengths:**

- Tri-BBAI is the first batched algorithm to achieve optimal sample complexity in an asymptotic setting
- Opt-BBAI is the first batched algorithm to achieve near-optimal sample and batch complexity in non-asymptotic setting
- Both algorithms are supported by valid theoretical analysis
- Batched solutions may bring benefits to real-world scenarios that cannot rely on sequential methods, like Track-and-Stop algorithms

**Weaknesses:**

- The paper dedicates a significant portion to the introduction and related works, only beginning to describe methodologies and algorithms on page 5. I suggest reducing the introductory section to allow more space for discussing comparisons with existing algorithms (Table 2) and for elaborating on the experimental results.
- The paragraph on the notation is partially useful, since it presents some notations that are never utilized in the main paper, but only in the appendix and may be employed to describe other unclear symbols.
- I found some statements a bit misleading. For example, at line 251 the period ”In this section, we introduce Opt-BBAI, which can attain the optimal sample and batch complexity in both asymptotic and non-asymptotic settings adaptively [...]” is in contrast with lines 255-256 where, always regarding Opt-BBAI, is correctly said that ”we can achieve asymptotic optimality and near non-asymptotic optimality adaptively[...].
- The claim at line 714 ”the sample complexity of Tri-BBAI and Opt-BBAI [...] is at most 2.6 times greater than Track and Stop when δ is very small” is not true, since in the second experiment for δ = 1 × 10−10 the ratio is evidently larger.

**Questions:**

- Reward distributions are assumed to belong to a single one-parameter exponential family, a common choice in literature. Are there any limitations in choosing different reward distributions that could affect the effectiveness of the presented approach?
- I have some doubts about the necessity of presenting both Tri-BBAI and Opt-BBAI in the paper. If Opt-BBAI achieves the same performances of Tri-BBAI in the asymptotic settings, why is it necessary to describe Tri-BBAI as well, considering that Opt-BBAI also addresses non-asymptotic settings? Could you explain and give some intuition about the need to
present Tri-BBAI?
- The choice of exactly three batches in the Tri-BBAI algorithm is not entirely clear. Additionally, I would like to ask the authors to provide some intuition regarding the optimal batch complexity of 2 for both Tri-BBAI and Opt-BBAI (shown in Table 1) compared to the structure of Tri-BBAI algorithm which employs at most 3 batches. What is the intuition behind these results? Why was a 2-batch algorithm not considered from the beginning?
- Experiments have been carried on crafted environments. Since authors argued about the benefits that their work would bring in real-world settings, I would expect to find an experiment on a such real-world scenario.

---

> ### Author Rebuttal · Authors · 2024-08-07
>
> We thank the reviewer for their valuable time and effort in providing detailed feedback on our work. In the following, we present answers to your suggestions (S) and questions (Q) point-by-point. We hope our response will fully address all of the reviewer's points.
>
> ---
> S1: Reduce the introductory section to allow more space for discussing comparisons with existing algorithms (Table 2) and for elaborating on the experimental results.
>
> A-S1: Thank you for this suggestion. In our revision, we will shorten the introduction and move the related work section to the last two sections before the acknowledgments. We will only include additional related work that was not previously introduced. The space saved will be used to discuss more comparisons with existing algorithms (in Table 2 of our manuscript) and present our experimental results.
>
> ---
>
> S2: The paragraph on the notation is partially useful.
>
> A-S2:  We agree. The paragraph on the nation will be moved to the appendix.
>
> ---
> S3: I found some statements a bit misleading. For example, at line 251 the period ”In this section, we introduce Opt-BBAI, which can attain the optimal sample and batch complexity in both asymptotic and non-asymptotic settings adaptively [...]” is in contrast with lines 255-256 where, always regarding Opt-BBAI, is correctly said that ”we can achieve asymptotic optimality and near non-asymptotic optimality adaptively.
>
> A-S3: Thank you for careful reading and pointing out this typo. We will revise the paper accordingly, i.e., claim that we adaptively attain the optimal sample and batch complexity in asymptotic settings and near-optimal sample and batch complexity in non-asymptotic settings.
>
> ---
>
> S4: The claim at line 714 ”the sample complexity of Tri-BBAI and Opt-BBAI [...] is at most 2.6 times greater than Track and Stop when δ is very small” is not true, since in the second experiment for $\delta = 1 × 10^{−10}$ the ratio is evidently larger.
>
> A-S4: Thanks for pointing out this. It should be 3.6.
>
> ---
>
> Q1: Reward distributions are assumed to belong to a single one-parameter exponential family, a common choice in literature. Are there any limitations in choosing different reward distributions that could affect the effectiveness of the presented approach?
>
> A-Q1: For any reward distribution that belongs to an exponential family, the algorithm satisfying Eq (1.3) is asymptotically optimal. Therefore, different reward distributions do not affect the asymptotic behavior of my algorithm, as it is asymptotically optimal.
>
> ---
> Q2: I have some doubts about the necessity of presenting both Tri-BBAI and Opt-BBAI in the paper. If Opt-BBAI achieves the same performances of Tri-BBAI in the asymptotic settings, why is it necessary to describe Tri-BBAI as well, considering that Opt-BBAI also addresses non-asymptotic settings? Could you explain and give some intuition about the need to present Tri-BBAI?
>
> A-Q2:  Our first algorithm introduces a method for designing an algorithm that achieves asymptotic optimality within constant batches.
> Our second algorithm presents a framework for integrating an asymptotically optimal algorithm with one that has near-optimal non-asymptotic sample complexity. This results in a new algorithm that is optimal or near-optimal in both asymptotic and non-asymptotic settings. Additionally, the second algorithm provides new techniques to ensure that the sample complexity does not depend on high probability events.
> The primary focuses of the two algorithms are distinct. Combining them into a single algorithm would result in a lengthy and complex structure, potentially making it difficult for the reader to grasp the core ideas.
>
> ---
>
> Q3: Intuition regarding the optimal batch complexity of 2 for both Tri-BBAI and Opt-BBAI (shown in Table 1) compared to the structure of Tri-BBAI algorithm which employs at most 3 batches. What is the intuition behind these results?
>
> A-Q3:  This is a typo. The batch complexity in Table 1 of my manuscript should be 3, and we will correct this accordingly.
>
> We provide an intuitive explanation of my algorithm by detailing the purpose of each step, specifically in Lines 181-184, 187-189, 209-213, and 217-219.
>
> Consider a two-armed bandit problem with a gap $\Delta$. Previous studies estimate $\Delta$ using exponentially decreasing values, such as $1/2$, $1/2^2$, etc., requiring at least $\log (1/\Delta)$ batches. In Tri-BBAI, we initially explore each arm $\sqrt{\log (1/\delta)}$ times. This initial exploration ensures that the true means of each arm is in a small range with a small failure probability of $p = 1/\log^2(1/\delta)$. We then use the estimated means to calculate the optimal sample size for each arm, as defined by asymptotic optimality (see Eq. 3.1). In the subsequent exploration phase, we sample each arm to the calculated optimal size and verify whether the best arm is identified with a probability of $1-\delta$.
>
>
> ---
> Q4: Real-world experiments.
>
> A-Q4: Obtaining real-world data presents significant challenges. In the literature on best arm identification that explores asymptotic optimality, experiments are typically conducted on synthetic data. We consider conducting real-world experiments to be an interesting direction for future work.

---

> > ### Author Response · Authors · 2024-08-12
> >
> > Dear Reviewer YCPj,
> >
> > We would like to know whether our responses have addressed your concerns. If there are any additional questions or areas that require clarification, please do not hesitate to let us know.
> >
> > Thank you.

---

> > > ### Comment · Reviewer_YCPj · 2024-08-13
> > >
> > > Thanks for the answers to my concerns. They are satisfactory.

---

### Author Rebuttal · Authors · 2024-08-07

We would like to express our gratitude to all the reviewers for your insightful comments and for recognizing the strengths of our work. We summarize these strengths as follows:
* Reviewer YCPj, Reviewer 3W3Z, and Reviewer UhZ8 all appreciated that our algorithm achieves the optimal sample and batch complexity in asymptotic settings and near-optimal sample and batch complexity in non-asymptotic settings. In particular, Reviewer 3W3Z states that these are nice results, and Reviewer UhZ8 states that the results are pretty strong.
* Reviewer YCPj and Reviewer 3W3Z valued the potential practical use of our algorithm. Reviewer YCPj mentions that batched solutions may benefit real-world scenarios that cannot rely on sequential methods, such as Track-and-Stop algorithms, and Reviewer 3W3Z notes that best arm identification under fixed confidence is a well-studied problem and running algorithms in batches has clear practical relevance.
* Reviewer UhZ8 appreciated our consideration of perspectives different from prior work, specifically: (1) asymptotic optimality of sample complexity, and (2) focusing on the expected number of rounds rather than treating it as a hard constraint.

We also received numerous valuable suggestions and have revised the paper accordingly. In particular, the following major changes have been made in our response and will be incorporated into our final version:
1.  As suggested by Reviewer vkzZ, we added two baselines. Due to space limitations, we presented the experimental results on the Normal dataset in the following Table 1 and the Uniform dataset in Table 2 of the attached PDF file. The results are consistent across both the Normal and Uniform datasets, and we will incorporate them in the final version.
2. As suggested by Reviewer YCPj, we shortened the introduction. The space saved will be used to discuss more comparisons with existing algorithms presented in Table 2 of our manuscript and present our main experimental results.



---
#### **Table 1:** Experimental results in terms of sample complexity, batch complexity and runtime under the normal mean rewards. The methods are ID-BAI, CollabTopM and our two algorithms. The number of arms is $n=10$. The experimental results are averaged over 1000 repetitions.
| Dataset | $\delta$ | Algorithm | Sample Complexity | Batch Complexity | Runtime (s) | Recall |
|---|---|---|---|---|---|---|
| **Normal** | **$1×10^{-1}$** | ID-BAI | 120427±0 | - | 0.145±0.012 | 100% |
| | | CollabTopM | 11834.089±2208.691 | 2.888±0.315 | 0.015±0.003 | 100% |
| | | Tri-BBAI | 334.49±32.80 | 3.89±0.32 | 1.08±0.16 | 98.3% |
| | | Opt-BBAI | 793.23±358.15 | 4.18±0.79 | 1.05±0.16 | 100% |
| | **$1×10^{-2}$** | ID-BAI | 146948±0 | - | 0.171±0.011 | 100% |
| | | CollabTopM | 16145.218±2255.120 | 2.933±0.250 | 0.021±0.004 | 100% |
| | | Tri-BBAI | 893.05±121.47 | 3.62±0.48 | 1.11±0.11 | 100% |
| | | Opt-BBAI | 1236.33±381.17 | 3.74±0.71 | 1.11±0.13 | 100% |
| | **$1×10^{-3}$**  | ID-BAI | 173478±0 | - | 0.204±0.013 | 100% |
| | | CollabTopM | 20414.549±2277.714 | 2.960±0.195 | 0.027±0.004 | 100% |
| | | Tri-BBAI | 1532.76±201.18 | 3.37±0.48 | 1.13±0.10 | 100% |
| | | Opt-BBAI | 1747.15±389.65 | 3.43±0.58 | 1.07±0.10 | 100% |
| | **$1×10^{-4}$** |  ID-BAI | 199999±0 | - | 0.236±0.015 | 100% |
| | | CollabTopM | 24673.111±2584.898 | 2.962±0.191 | 0.032±0.005 | 100% |
| | | Tri-BBAI | 2141.11±282.37 | 3.20±0.40 | 1.13±0.08 | 100% |
| | | Opt-BBAI | 2263.174±405.72 | 3.19±0.42 | 1.06±0.08 | 100% |
| | **$1×10^{-5}$** | ID-BAI | 226528±0 | - | 0.269±0.044 | 100% |
| | | CollabTopM | 29081.141±2323.105 | 2.982±0.132 | 0.039±0.005 | 100% |
| | | Tri-BBAI | 2838.36±353.12 | 3.09±0.28 | 1.14±0.08 | 100% |
| | | Opt-BBAI | 2881.61±430.05 | 3.08±0.27 | 1.09±0.09 | 100% |
| | **$1×10^{-6}$** | ID-BAI | 253049±0 | - | 0.312±0.038 | 100% |
| | | CollabTopM | 33386.52±2133.062 | 2.987±0.113 | 0.043±0.005 | 100% |
| | | Tri-BBAI | 3516.52±467.48 | 3.03±0.18 | 1.14±0.08 | 100% |
| | | Opt-BBAI | 3556.59±477.73 | 3.04±0.20 | 1.15±0.15 | 100% |
| | **$1×10^{-7}$** | ID-BAI | 279579±0 | - | 0.336±0.019 | 100% |
| | | CollabTopM | 37499.802±2413.418 | 2.986±0.117 | 0.048±0.005 | 100% |
| | | Tri-BBAI | 4218.39±523.49 | 3.01±0.09 | 1.15±0.08 | 100% |
| | | Opt-BBAI | 4220.78±523.11 | 3.02±0.13 | 1.16±0.13 | 100% |
| | **$1×10^{-8}$** | ID-BAI | 306108±0 | - | 0.362±0.018 | 100% |
| | | CollabTopM | 41924.698±2206.559 | 2.992±0.089 | 0.055±0.005 | 100% |
| | | Tri-BBAI | 4912.64±613.69 | 3.01±0.08 | 1.15±0.07 | 100% |
| | | Opt-BBAI | 4940.68±650.75 | 3.00±0.06 | 1.12±0.10 | 100% |
| | **$1×10^{-9}$** | ID-BAI | 332628±0 | - | 0.393±0.017 | 100% |
| | | CollabTopM | 46244.241±1816.041 | 2.997±0.054 | 0.059±0.005 | 100% |
| | | Tri-BBAI | 5637.36±743.53 | 3.00±0.05 | 1.14±0.08 | 100% |
| | | Opt-BBAI | 5661.51±740.74 | 3.00±0.05 | 1.03±0.06 | 100% |
| | **$1×10^{-10}$** | ID-BAI | 359158±0 | - | 0.423±0.015 | 100% |
| | | CollabTopM | 50556.41±1822.998 | 2.997±0.054 | 0.067±0.007 | 100% |
| | | Tri-BBAI | 6356.78±787.53 | 3.00±0.03 | 1.30±0.16 | 100% |
| | | Opt-BBAI | 6355.28±793.45 | 3.00±0.03 | 1.03±0.05 | 100% |

---

### Decision · Program_Chairs · 2024-09-25

**Decision:**

Accept (poster)

**Comment:**

This paper discusses the best arm identification in the bandit problem under the batched setting. Two algorithms are proposed for this problem. The first one achieves the asymptotic optimality with respect to the error probability within almost three batches, and the second algorithm achieves the optimality in both the error probability and other parameters with the number of batches depending on the gap parameter. Results of experiments are given in the appendix to verify the performance of the algorithms.

The main strength of the paper is that the proposed algorithms achieve good sample complexities with a theoretically and empirically very small number of batches in various settings. On the other hand, there are several concerns on the writing/structure of the paper and comparison with other algorithms. The authors gave a plan of the revision to address these concerns and some additional results of the experiments are shown in the rebuttal. I judged that they are reasonable and I determined to recommend acceptance, but I expect that the authors very carefully revise the paper so that it falls within a minor revision while addressing all the above concerns.

Another criticism on the paper is that the asymptotics in $\delta$ is not very important. Though I partly agree that very small $\delta$ is practically not realistic, I think it is still significant that the optimality in $\delta$ is achievable jointly with the optimality in other parameters unless the criticism is accompanied with some support on the triviality of this combination. Still, I expect that the authors explain the significance on this point more clearly in the final version.

Following is my own comment to the paper:
- On Algorithm 1: in Line 15, $t$ and $\tau_q$ are opposite? I also got a similar concern on the unused notation as YCPj like $\tau_q$, and I hope that the authors very carefully check them.
- My concern is that, not only the sample complexity is 2-3 times worse than the Track-and-Stop algorithm even for $\delta=10^{-10}$, its slope with respect to $\log 1/\delta$ also seems 4-5 times worse even though $\alpha=1.001$ (very close to 1) is used in the experiments. This seems to mean that the asymptotic behavior in e.g. Theorem 3.1 does not appear even in this extremely small $\delta$. I think there should be some discussion on this point in relation with the concern by Uhz8. Related to this, why is $\alpha=1.001$ used in the experiments rather than $\alpha=1+1/\log \delta^{-1}$ suggested below Theorem 3.1? I don't see the necessity of the use of $\alpha$ very close to 1 if the asymptotic behavior cannot be observed.